# Mobile Sleep Lab: Comparison of polysomnographic parameters with a conventional sleep laboratory

Chihiro Suzuki[1], Yoko Suzuki[1], Takashi Abe[1]*, Takashi Kanbayashi[1,2], Shoji Fukusumi[1], Toshio Kokubo[1,3], Isamu Takahara[1,4,5], Masashi Yanagisawa[1,3,6]*

1 International Institute for Integrative Sleep Medicine (WPI-IIIS), University of Tsukuba, Tsukuba, Ibaraki, Japan, 2 Ibaraki Prefectural Medical Center of Psychiatry, Kasama, Ibaraki, Japan, 3 S'UIMIN Inc., Shibuya, Japan, 4 Center for Cybernics Research, University of Tsukuba, Tsukuba, Ibaraki, Japan, 5 CYBERDYNE, Inc., Tsukuba, Ibaraki, Japan, 6 R&D Center for Frontiers of MIRAI in Policy and Technology (F-MIRAI), University of Tsukuba, Tsukuba, Ibaraki, Japan

☉ These authors contributed equally to this work.
* abe.takashi.gp@u.tsukuba.ac.jp (TA); yanagisawa.masa.fu@u.tsukuba.ac.jp (MY)

## Abstract

In remote areas, visiting a laboratory for sleep testing is inconvenient. We, therefore, developed a Mobile Sleep Lab in a bus powered by fuel cells with two sleep measurement chambers. As the environment in the bus could affect sleep, we examined whether sleep testing in the Mobile Sleep Lab was as feasible as in a conventional sleep laboratory (Human Sleep Lab). We tested 15 healthy adults for four nights using polysomnography (the first two nights at the Human Sleep Lab or Mobile Sleep Lab with a switch to the other facility for the next two nights). Sleep variables of the four measurements were used to assess the discrepancy of different places or different nights. No significant differences were found between the laboratories other than the percentage of total sleep time in stage N3. Next, we analyzed the intraclass correlation coefficient to evaluate the test-retest reliability. The intraclass correlation coefficient between these two measurements: the Human Sleep Lab and Mobile Sleep Lab showed similar reliability for the same sleep variables. The intraclass correlation coefficient revealed that several sleep indexes, such as total sleep time, sleep efficiency, wake after sleep onset, percentage of stage N1, and stage R latency, showed poor reliabilities (<0.5) based on Koo and Li's criteria. In contrast, the percentage of stage N3 showed moderate (0.5–0.75) or good (0.75–0.9) reliabilities. As almost all sleep variables showed no difference and same level of test-retest reliability between the Mobile Sleep Lab and Human Sleep Lab, the Mobile Sleep Lab might be suitable for conducting polysomnography as a conventional sleep laboratory. The reduction in N3 in the Mobile Sleep Lab should be scrutinized in the larger sample, including sleep disorders. Practical application of the Mobile Sleep Lab can transform sleep medicine in remote areas.

**Data Availability Statement:** The raw polysomnographic data are available without restriction on Zenodo (https://doi.org/10.5281/zenodo.14020530).

**Funding:** The hydrogen-fueled bus used in the study was rented free of charge from Toyota Motor Cooperation. Dr. Yanagisawa's work has been funded by the "Social Application of Mobility Innovation and Future Social Engineering Research Phase IV," a joint research project between Toyota Motor Corporation and the University of Tsukuba; and by the MEXT, WPI program; and AMED under Grant Number JP21zf0127005. Dr. Takahara was a former employee of Toyota Motor Cooperation, and the general manager in charge of the Future Development Office, Frontier Research Center, Toyota Motor Corporation. The other authors declare no potential conflict of interest. This does not alter our adherence to PLOS ONE policies on sharing data and materials. The funders had no role in study design, data collection and analysis, decision to publish, or preparation of the manuscript.

**Competing interests:** I have read the journal's policy and the authors of this manuscript have the following competing interests: The hydrogen-fueled bus used in the study was rented free of charge from Toyota Motor Cooperation. Dr. Yanagisawa's work has been funded by the "Social Application of Mobility Innovation and Future Social Engineering Research Phase IV," a joint research project between Toyota Motor Corporation and the University of Tsukuba; and by the MEXT, WPI program; and AMED under Grant Number JP21zf0127005. Dr. Takahara was a former employee of Toyota Motor Cooperation, and the general manager in charge of the Future Development Office, Frontier Research Center, Toyota Motor Corporation. The other authors declare no potential conflict of interest. This does not alter our adherence to PLOS ONE policies on sharing data and materials.

## Introduction

High-precision sleep testing requires visiting a sleep laboratory and undergoing polysomnography (PSG) [1]. However, for those residing in remote mountainous regions or islands, this is unfeasible due to the lack sleep laboratories, specifically in the 416 inhabited islands of Japan [2]. Therefore, portable monitoring and out-of-center sleep testing have been developed to solve this problem.

Portable electroencephalography (EEG) enables sleep testing without needing to visit a sleep laboratory, due to the various recently-introduced advanced EEG devices [3–10]. Compared with PSG, portable EEG is less expensive, uses fewer electrodes, and is easier to apply [3]. The agreement rate based on sleep stage scoring between PSG and portable EEG has reached 83.5% [3]; thus, portable EEG can be used to accurately measure sleep EEG. Several clinical investigative studies have utilized portable EEGs in patients with insomnia and rapid eye movement (REM) sleep behavior disorder (RBD) [7–9]. Recently, combining a portable EEG with a pulse oximeter has enabled the screening for obstructive sleep apnea (OSA) [10]. Thus, portable EEG has the advantage of accurately being able to assess sleep stages, and the sleep disorders that can be diagnosed with portable EEG will increase.

Paving the way for remote medical care, home sleep testing (HST/unattended, limited-channel sleep testing) has become increasingly popular worldwide. HST can perform breathing and leg movement tests that are difficult to perform with a portable EEG alone. The advantage of HST is the convenience of evaluating patients with OSA at home [11] and reduced costs compared with PSG at a hospital [12]. Previous studies reported the comparable accuracies between laboratory PSG and HST in determining respiratory events and poor PSG signal rates; furthermore, HST showed better sleep quality than laboratory PSG as it overcame the first night effect (FNE), or the disturbance of sleep structure when sleeping in a new environment [1, 13–15]. HST provides more biological information than portable EEG to diagnose sleep disorders and has advantages over laboratory PSG.

The need for remote sleep testing remains unmet despite the advancing of HST technology. PSG performed by a sleep technologist is necessary for a high-precision examination of sleep disorders. At present, HST is limited to specific sleep disorders, and the patients that most benefit from HST are those with a moderate-to-high probability of OSA and no comorbidities that can degrade the study's accuracy [16]. In addition, the multiple sleep latency test (MSLT) and the maintenance of wakefulness test (MWT) require the technician to score the sleep stages in real-time based on the PSG. Furthermore, high-precision PSG is needed to control the environment (light, sound, temperature, and humidity). Therefore, to solve these issues, we aimed to create a reproducible mobile environment equivalent to that of a regular sleep laboratory for individuals with actual or suspected sleep disorders where laboratory testing may be unfeasible.

Our goal is to develop a society where PSG examination is accessible to everyone, no matter where they are situated. To this end, we developed a mobile sleep laboratory (Fig 1), called the "Mobile Sleep Lab" (MSL). The current literature lacks any searchable research on the development of a mobile sleep laboratory. The MSL enables PSG examination at any location in the presence of a sleep technologist. To supply the electric power necessary for air conditioning, lighting, monitoring, and PSG, the engine of a conventional bus was considered; however, it generates considerable noise and vibration. Nevertheless, a hydrogen fuel-cell system can be utilized to supply electric power quietly. Therefore, we installed the MSL in a hydrogen-fueled bus to prevent sleep disturbances caused by the engine.

Nevertheless, environmental factors inside such a bus (e.g., electrical equipment noise and limited space) could interfere with the sleep of the participant undergoing PSG even when the

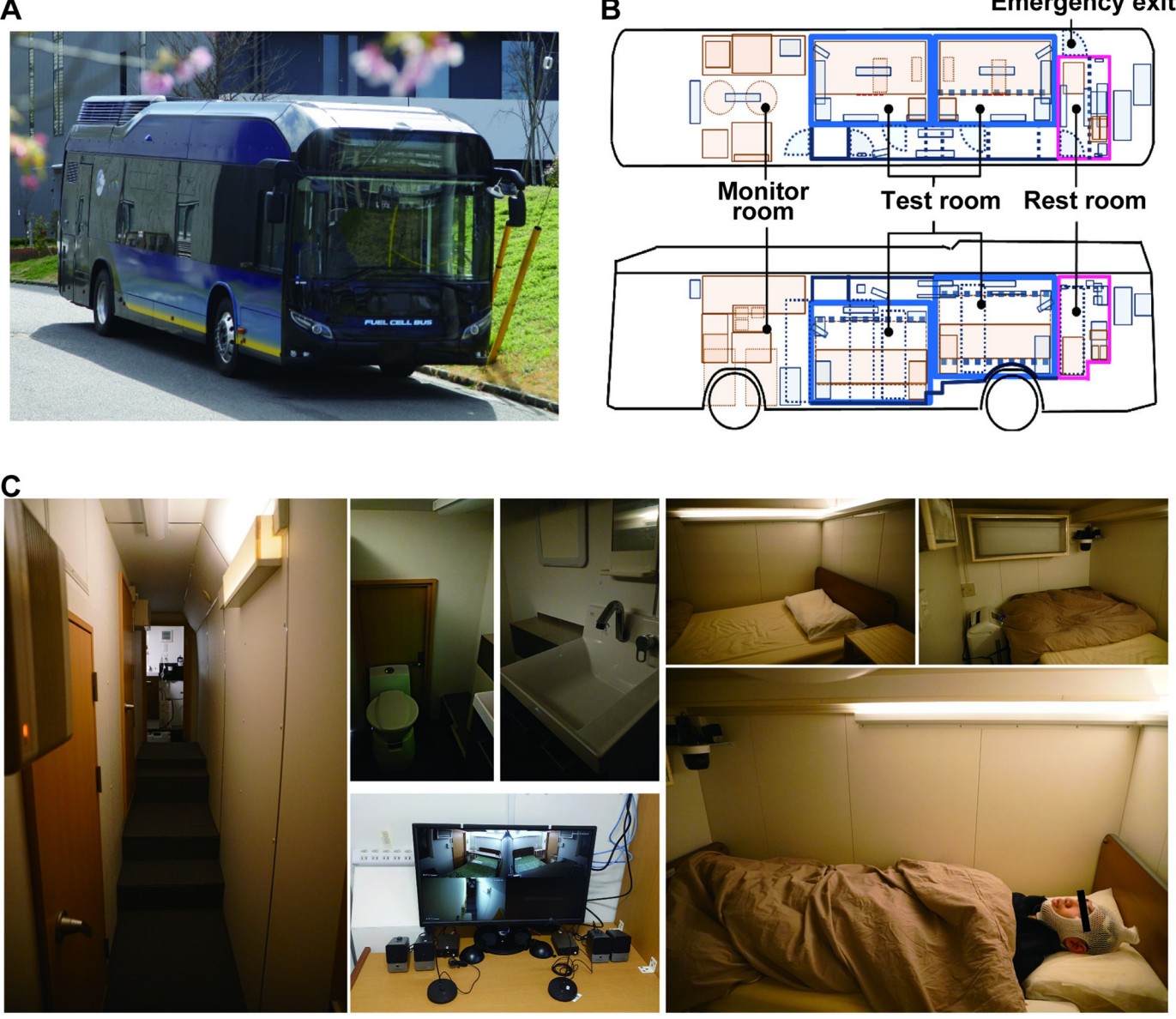

**Fig 1. The Mobile Sleep Lab.** (A) External appearance, (B) design (the upper half shows the top view, and the lower half shows the left side of the bus, relative to the direction of travel); (C) the corridor, restroom, monitor room, and sleep chamber of the Mobile Sleep Lab. The logo of the International Institute for Integrative Sleep Medicine (WPI-IIIS) in this Fig 1A is used with permission from the institute. The photograph (Fig 1A), including the exterior design of the Mobile Sleep Lab has been used with permission from the designer, Tadanobu Hara. Permission to use the photographs and images (Fig 1A–1C) of the fuel cell bus was obtained from Toyota Motor Corporation [17].

hydrogen power is not in use. Therefore, we aimed to examine if the sleep-variable measurements made in an MSL are equivalent to those made in a conventional sleep lab (Human Sleep Lab [HSL]). For the first step, we investigated whether the MSL and HSL were equivalently capable of staging sleep for a polysomnographic study in adults, in reference to the American Academy of Sleep Medicine (AASM) criteria [18].

## Methods

### Participants

Following verbal and written explanations of the study, written informed consent was obtained from all participants. This study was approved by the Tsukuba Clinical Research and Development Organization (Ethics Review No. R01-392) and was pre-registered in the University hospital Medical Information Network Clinical Trials Registry (UMIN-CTR; UMIN000040114). This study has been conducted based on the tenets of the Declaration of Helsinki.

Sixteen healthy adults were recruited through internet bulletin boards and the laboratory website. The recruitment period was from April 23, 2020 to June 5, 2020. Inclusion criteria were as follows: age 20–59 years, literacy in Japanese, those who can stay and sleep overnight in both the HSL and MSL, not currently under treatment for sleep disorders, and no previous experience with PSG. Exclusion criteria were: (i) irregular lifestyle with bedtimes outside the range of 21:00 to 1:00 h, waking up before or after 6:00 to 9:00 h, or more or less than 7–9 h of sleep each night; (ii) body mass index (BMI) of <18.5 or $\geq$25 kg/m$^2$; (iii) a history of night-shift work after 22:00 h; (iv) travel history in the last three months to a country with more than three hours' time difference with Japan; (v) habitual alcohol intake ($\geq$40 g of alcohol at least twice a week); (vi) smoking habit; (vii) caffeine intake of $\geq$300 mg per day; (viii) acute disease; (ix) claustrophobia; and (x) pregnant/breastfeeding. The Pittsburgh Sleep Quality Index (PSQI) was used to assess self-reported sleep quality [19, 20]. Poor sleep quality was defined as a PSQI score of $\geq$6 [19]. Participants with PSQI scores of $\geq$5.5 were excluded. To determine the participants' chronotype, we used the Morningness-Eveningness Questionnaire (MEQ) [21, 22]. Participants who scored $\leq$30 points (definitely morning type) or $\geq$70 points (definitely evening type) [21, 22] were excluded. We recruited 25 participants; four withdrew consent, one had a history of PSG and another four were excluded due to their ineligible BMI (S1 Fig). One participant withdrew from the experiment after completing the first measurement (S1 Fig). The data of 15 participants (five female) aged 21.7±1.6 [20.0–26.0] years (mean±standard deviation [range]) were included in the final analysis (Table 1 and S1 Fig). Demographic characteristics of the participants are shown in Table 1. Fifteen patients were finally analyzed as one participant did not complete the study.

**Table 1. Participant characteristics.**

|  | Number | Percentage (%) | Mean±SD | Range |
|---|---|---|---|---|
| **Sex** |  |  |  |  |
| Male | 10 | 66.7 |  |  |
| Female | 5 | 33.3 |  |  |
| **Age (years)** |  |  | 21.7±1.6 | 20.0–26.0 |
| **BMI (kg/m$^2$)** |  |  | 21.0±1.5 | 18.7–23.5 |
| **MEQ score** |  |  | 51.0±6.3 | 39.0–61.0 |
| **MEQ type** |  |  |  |  |
| Moderate Morning | 1 | 6.7 |  |  |
| Intermediate | 12 | 80.0 |  |  |
| Moderate Evening | 2 | 13.3 |  |  |
| **PSQI score** |  |  | 3.3±1.4 | 0.0–5.0 |

BMI, body mass index; SD, standard deviation; MEQ, Morningness-Eveningness Questionnaire; PSQI, Pittsburgh Sleep Quality Index.

## Procedures

A randomized, open-label crossover design was employed. The study design and protocol are presented in Fig 2. We measured the sleep of participants for four nights in each of the two laboratories. The participants were randomly allocated to two groups with equal numbers of participants (n = 8 each). The first group underwent sleep measurement at the HSL for the first two nights and at the MSL for the next two nights; this order was reversed for the second group. Since one participant withdrew, seven underwent PSG first at the HSL, followed by the MSL. In contrast, eight underwent PSG in the order of MSL, followed by HSL.

The participants' sleep habits were assessed and controlled three days before the experiment. The participants were instructed to maintain their regular lifestyles, go to bed at 21:00–1:00, wake up at 6:00–9:00, and have 7–9 h of sleep. Their sleep-wake cycle was recorded during the three days using the consumer sleep tracker Fitbit (Fitbit Charge 3; Fitbit, San Francisco, CA) based on acceleration and pulse-rate recordings by a wrist-mounted device, which has been used in previous sleep research [23]. On the experiment days, participants were asked to refrain from napping, excessive exercise, alcohol consumption, and medication; from the afternoon of each day, they were asked to abstain from taking medicines that affect sleep, such as antihistamines for hay fever, or caffeine. The study included only healthy participants and

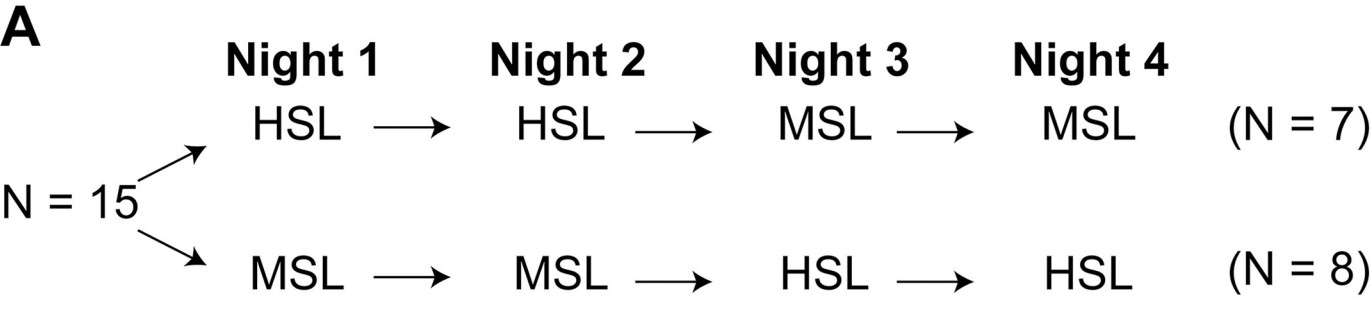

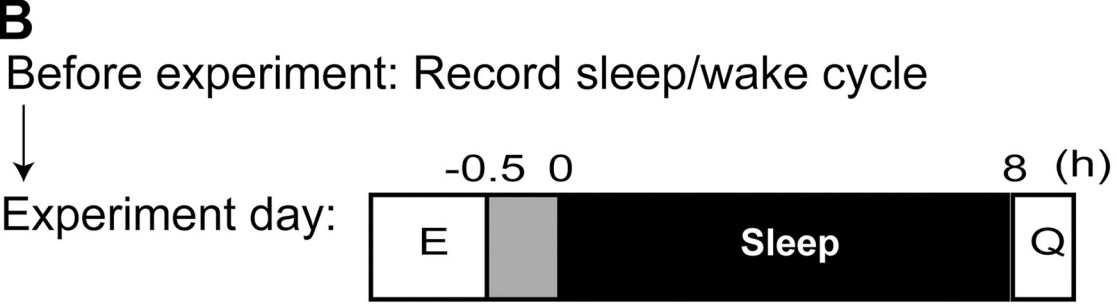

**Fig 2. Study design and protocol.** (A) Study design. The study employed a randomized, open-label, crossover design. Sleep was measured in the laboratory over four consecutive nights. Half of the participants underwent polysomnography (PSG) at the Human Sleep Lab (HSL) for the first two nights, and at the Mobile Sleep Lab (MSL) for the following two nights. The participants were randomly allocated to two groups. The first group (n = 7) underwent sleep measurement at the HSL for the first two nights, and at the MSL for the next two nights; this sequence was reversed for the second group (n = 8). (B) Protocol. The participants were instructed to maintain their regular lifestyles during the three days preceding the experiment, and their sleep-wake cycles were recorded using Fitbit Charge 3 and a sleep log. PSG electrodes were attached during lab assessments. All electronic devices, including smartphones and personal computers, were turned off from 30 min before lights out until lights on. Light intensity was limited to <50 lux for 30 min before lights out (gray bar). Each participant was permitted eight hours of sleep of the total recording time (TRT, black bar). Lights-out time was adjusted to match the participants' typical bedtime, and the lights-out and lights-on times were consistent throughout the experiment. The time course of the experiment has been given. (0), lights out, (E) PSG electrodes attached; (Q) questionnaire (Oguri-Shirakawa-Azumi Sleep Inventory, Middle-age and Aged version) responses recorded.

excluded those with diseases; thus, those with underlying medical conditions requiring routine medication were not included. Additionally, electronic devices (e.g., smartphones and personal computers) were turned off 30 min before lights out until after waking up, and lighting in the room was kept at <50 lux. Each participant was allowed to sleep for 8 h each night. The participants adhered to their usual bedtime, and the lights-out and lights-on times were constant throughout the experiment.

## Laboratory room

Experiments were conducted in two laboratories: the HSL and MSL. The internal dimensions of the HSL were 3,022 mm × 1,702 mm × 2,110 mm (length × width × height) with a floor area of 5.1 m$^2$. The internal dimensions of the MSL were 2,245 mm × 1,450 mm × 1,627.5 mm, and the floor area was 3.3 m$^2$.

The sound insulation was D-40 dB and conformed to ISO 717–1 [24]. A monitoring room and living room were provided for the experimenter, and video surveillance of the measurement room was available (Fig 1). An external power supply powered the bus. Using an air conditioner (HSL: F22VTES-W, MSL: C22VTSXV-W, Daikin Industries, Ltd., Osaka, Japan), each testing room was set to a temperature that the participants found comfortable. As the humidity suitable for sleep is 40–60% [25], a dehumidifier was used to control the humidity (MCZ70-W, Daikin Industries, Ltd.). A thermo-hygrometer (Testo 175H1 temperature and humidity data logger, Testo SE & Co. KGaA., Titisee-Neustadt, Germany), sound level meter (SL-1373SD, CUSTOM Co., Tokyo, Japan), and vibrometer (VB-8206SD, MOTHERTOOL Co., Ltd., Nagano, Japan) were used to confirm the environmental parameters nightly. We analyzed the temperature, humidity, sound, and vibration, and the coefficient of variation (CV) of each parameter only during the 8 h of nighttime sleep. We calculated the average temperature, humidity, and sound in each hour separately.

## PSG measures

PSG using the Polymate Pro MP6100 (Miyuki Giken, Tokyo, Japan) was performed following the AASM criteria [18] and consisted of EEG, electromyography (EMG), electrooculography (EOG), and electrocardiography (ECG). EEG electrodes were placed based on the International 10–20 System [26]. We recorded eight EEG electrodes on F3, F4, C3, C4, O1, O2, M1, and M2 [18]. EOG electrodes were placed at two locations: 1 cm below and 1 cm lateral to the outer canthus of the left eye and 1 cm above and 1 cm lateral to the outer canthus of the right eye [18]. EMG electrodes were placed at three locations: 1 cm above the lower border of the mandibular midline; 2 cm below the lower border and 2 cm towards the right from the mandibular midline; and 2 cm below the lower border and 2 cm towards the left from the mandibular midline [18]. We used a single modified ECG Lead II for ECG recordings [18]. EEG, EOG, EMG, and ECG were sampled at 500 Hz and recorded with a high frequency of 160 Hz and a low frequency of direct current (DC). In this study, PSG was performed in the MSL by stopping the bus at the same location across the experiment. The output from the Polymate Pro was in the European Data Format (EDF), setting the high-pass filter to 0.03 Hz and the low-pass filter to 200 Hz [27]. We used the EDF files for PSG analysis and EEG power spectral analysis.

## PSG analysis

Sleep stages were determined using the AASM criteria [18]. A registered polysomnographic technologist (YS), blinded to identification and conditions, scored the PSG using Sleepware G3 (version 3.9.4, Respironics, Inc., Murrysville, PA). In addition, the following sleep

parameters were calculated: total sleep time (TST), sleep latency (SL), sleep efficiency (SE), wake after sleep onset (WASO), percentage of TST in each stage (%N1, %N2, %N3, and %R), non-rapid eye movement (NREM) sleep (N1, N2, and N3) latency from lights out, stage R latency (from sleep onset), and arousal index (ArI). ArI was calculated by arousal numbers/ TST×60 [18].

## EEG power spectral analysis

EEG data were sampled at 500 Hz and filtered using a low-frequency filter at 0.3 Hz and high-frequency at 35 Hz. We used MATLAB (MATLAB 2020b, MathWorks, Natick, MA) for the analysis. The EEG was visually observed; body movements and sweating were tagged as artifacts, and epochs containing artifacts were excluded from the analysis. The 4-s EEG recordings were fast Fourier transformed for the F3-M2, F4-M1, C3-M2, C4-M1, O1-M2, and O2-M1 channels after eliminating the DC component and applying a Hanning window. Ten power spectra of 4-s epochs with a 1-s overlap were assigned to a 30-s epoch. The slow oscillations (SO; 0.5–1.0 Hz), slow-wave activity (SWA; 0.5–4.0 Hz), delta-wave activity (1.0–4.0 Hz), theta-wave activity (4.0–8.0 Hz), alpha-wave activity (8.0–12.0 Hz), sigma-wave activity (12.0–16.0 Hz) and beta-wave activity (16.0–30.0 Hz) [28–30] were calculated for each 4-s epoch and averaged over ten epochs after natural log transformation. Thus, the average SWA of NREM (N2, N3) and the sum of SWAs for the 30-s epochs recorded during the night, i.e., the slow-wave energy (SWE), were obtained. The average of SWA was calculated only in N3. The average of SO, delta-wave activity, theta-wave activity, alpha-wave activity, and beta-wave activity was also calculated during NREM.

## Assessment of sleep perception

Self-reported sleep perception was investigated after waking using the Oguri-Shirakawa-Azumi Sleep Inventory, Middle-age and Aged version (OSA-MA), based on five factors: "sleepiness on rising," "initiation and maintenance of sleep," "frequent dreaming," "refreshness," and "sleep length." The responses comprised a closed choice of four options. The scores' polarity indicated high scores with good sleep. Standardized scores were used in the analysis [31].

## Statistical analysis

The normality of the data was tested using the Shapiro–Wilk test; Box-Cox transformations were performed for indicators for which normality was not confirmed and for which all values were >0. Indicators for which normality was confirmed in the raw data or after the Box-Cox transformation were analyzed using a linear mixed model; if normality was not confirmed after the Box-Cox transformation, they were analyzed using a non-parametric test.

We analyzed the following using a linear mixed model: sleep duration on Fitbit for three days of the sleep habits assessment period, room temperature, CV of temperature, CV of humidity, CV of sound level, TST, SE, SL, WASO, %N1, %N2, %N3, %R, N1, N2, N3 latency, ArI, SWA, SWE, and "initiation and maintenance of sleep" in the OSA-MA. For Fitbit sleep duration during the three-day assessment period, fixed effects were analyzed using the order of measurement (HSL→MSL/MSL→HSL), pre-experiment (days before the experiment), place (HSL/MSL), time (number of measurements at each laboratory: first/second night), and interactions among pre-experiment, place, and time. The repeated factor was days in the sleep habits assessment period (the first day of a total of 12 days was defined as one, and the last day as 12); each participant was defined as a block in the covariance structure. For the sleep variables, EEG power spectral and "initiation and maintenance of sleep" in the OSA-MA, the fixed effects were the order of measurement (HSL→MSL/MSL→HSL), place (HSL/MSL), timepoint

(number of measurements at each laboratory: first/second night), and interactions of place and timepoint. The repeated factor was night (first to fourth night), with each participant a block in the covariance structure. The linear mixed model was analyzed using *PROC MIXED* (SAS Version 9.4, SAS Institute, Cary, NC), with unstructured, autoregressive, and compound symmetry for covariance structures. Kenward–Roger's method was used for the degree-of-freedom calculations. Among the models that met the convergence criteria and had a positive-definite final Hessian matrix, the model with the minimum Akaike's information criterion was selected.

Friedman tests were conducted using four levels (place × time) for "sleepiness on rising," "frequent dreaming," "refreshness," and "sleep length" in the OSA-MA; humidity; sound level; vibration level; and stage-R latency; theta-wave activity in F3-M1 and C3-M1; beta-wave activity in F4-M1 and C4-M1 in the PSG (*PROC FREQ*, SAS Version 9.4). For the post-hoc analyses, Wilcoxon's signed rank-sum tests with Bonferroni correction were used for multiple comparisons, with *p*-values corrected by multiplying by the number of tests (*PROC UNIVARIATE*, SAS Version 9.4).

We calculated the effect sizes as Cohen's *d* (S1 Table) [32, 33]. For indicators with parametric tests, the least-squares mean and standard error were used. For indices with non-parametric tests, the raw data were used for calculation.

The sleep variables from the measurements on the first and second nights in the HSL, the first and second nights in the MSL, the first nights in the HSL and MSL, the second nights in the HSL and MSL, and the third and fourth nights (HSL/MSL) were used to calculate the intraclass correlation coefficient (ICC) for evaluating the test-retest reliability. The ICC (two-way mixed effects, absolute agreement, and single measurement) was judged as poor at $<0.5$, moderate at 0.5–0.75, good at 0.75–0.9, or excellent at $>0.9$ [34]. The ICCs were analyzed using SPSS Statistics Version 25.0 (IBM, Armonk, NY). The significance level was set at $p < 0.05$.

To assess the equivalence of the obtained sleep variables, we complementarily used the bio-equivalence determination method according to generic drug guidelines [35]. Equivalence was analyzed for the sleep variables obtained from the measurements in the "HSL on the first and second nights," "MSL on the first and second nights," "HSL and MSL on the first night," "HSL and MSL on the second night," and "HSL or MSL in the third and fourth nights." For the equivalence test, 90% confidence intervals (CIs) of differences in means and ratios of the values of the sleep variables were obtained. Differences in means and the 90% CIs of the differences were analyzed using *PROC MIXED*, a procedure in SAS used for analyzing linear mixed models. The ratios of the values were calculated as follows:

1. the difference in means between HSL measurements on the first and second nights/ mean of HSL measurements on the first night,

2. the difference in means between MSL measurements on the first and second nights/ mean of MSL measurements on the first night,

3. the difference in means between HSL and MSL measurements on the first night/ mean of HSL measurements on the first night,

4. the difference of means between HSL and MSL measurements on the second night/ mean of HSL measurements on the second night, and

5. the difference between the means of HSL and MSL measurements on the third and fourth nights/ mean of HSL measurements on the third night.

We judged the values as equivalent when the 90% CI of the difference in the means of the sleep variables was within ±20% of the ratio of the differences [35].

As we detected a significant decrease in %N3 and OSA-MA factor 2, "initiation and maintenance of sleep" in MSL compared to HSL, we calculated the correlation between the %N3 and OSA-MA using Pearson's correlation coefficient.

Hourly average was determined for temperature, humidity, and sound level and compared using SPSS Statistics Version 25.0. The Friedman test was used to see if there was any difference in whole-night variation in HSL and MSL. The Wilcoxon signed-rank test was used for post-hoc tests. For multiple comparisons, Bonferroni correction was used. As we found a significant increase in averaged sound in the first hour of MSL recording, we calculated the correlation between the %N3 and sound level in each hour from N = 472 data. Data intervals were removed where sleep was not occurring and sound level failed to be measured. Spearman's correlation coefficient was used to evaluate whether the %N3 decrement was related to sound level. As in the second night of MSL, the average hourly sound level at 1 h after lights out was higher than at 4 and 6–8 h, so we focused on the correlation between %N3 and sound level during the first hour. We predicted that the increase in sound level would be related to the decrease in %N3 in the MSL. The first and second nights were combined with the corresponding HSL and MSL to obtain the respective correlation coefficients between %N3 and sound level.

## Results

### Sleep duration prior to the experiment

Using the assigned Fitbit, sleep duration was evaluated for the three nights before the experiment and the average of the three days was calculated. Sleep duration for one, two, and three days before the experiment and the average sleep duration time over the three days are shown in S2 Fig. Differences were found in the sleep duration over the three days ($F_{1, 141} = 6.53$, $p = 0.012$; S2 Table), with significantly longer sleep on the first night than on the second night, each recorded in either the HSL or MSL ($t_{141} = 2.56$, $p = 0.012$, d = 0.51).

### Environmental measurements during the PSG period

No significant differences in temperature, sound levels, or vibration levels existed between sleep laboratories (S3A, S3C and S3D Fig, S3 and S4 Tables). Regarding humidity, an effect of the four levels of place × time ($F_{(3)} = 37.00$, $p < 0.0001$; S4 Table) was observed. Humidity was significantly higher in the HSL than in the MSL during the first and second nights (first nights in the HSL and MSL: S = −60, $p < 0.0001$, d = 1.71; second nights in the HSL and MSL: S = −60, $p < 0.0001$, d = 1.35; S3B Fig). There were main effects of place in CVs on temperature, humidity, and sound between the two laboratories ($F_{1, 14} = 6.96$, $p = 0.020$, $F_{1, 42} = 51.20$, $p < 0.0001$, $F_{1, 42} = 14.05$, $p < 0.001$, respectively; S3 Table). CVs of temperature and humidity showed significant increases in the MSL compared with the HSL ($t_{14} = −2.64$, $p = 0.020$, $t_{42} = −7.16$, $p < 0.0001$, respectively). On the contrary, the CV of sound level significantly decreased in the MSL compared to the HSL ($t_{42} = 3.75$, $p < 0.001$).

The average was calculated every hour to examine the effects of changes in environmental measurements on sleep over time (S4 Fig). In temperature, a main effect of time was found for MSL (first night: $F_{(7)} = 25.03$, $p < 0.001$; second night: $F_{(7)} = 17.5$, $p = 0.014$). No significant differences by time for both MSL nights were found in post-hoc tests (all $ps > 0.05$; S4A Fig). In humidity, a main effect of time was found for HSL (first night: $F_{(7)} = 45.4$, $p < 0.0001$; second night: $F_{(7)} = 22.1$, $p = 0.002$). Post-hoc tests revealed a significant increase in humidity at hour 7 versus hour 3 for the first night of HSL ($Z = −3.351$, $p = 0.023$; S4B Fig). There were no significant differences in post-hoc tests on the second night at the HSL (all $ps > 0.05$). In sound, a main effect of time was found for MSL (first night: $F_{(7)} = 22.05$, $p = 0.002$; second

night: $F_{(7)} = 45.49$, $p$ <0.0001). MSL first night post-hoc test showed no significant differences (all $ps$ >0.05). There were significant differences for the second night of MSL for 1 vs. 4 ($Z = -3.12$, $p = 0.049$), for 1 vs. 6 ($Z = -3.29$, $p = 0.028$), for 1 vs. 7 ($Z = -3.35$, $p = 0.023$) and 1 vs. 8 ($Z = -3.35$, $p = 0.023$), which indicated an increment in the sound level at the first hour compared within 4 and 6–8 hours.

To evaluate whether sound level increment in the first hour related to %N3 decrement in the MSL, correlations between hourly averaged sound level and hourly averaged %N3 were evaluated for the HSL and MSL. No significant correlation was found between sound levels in the HSL and MSL, respectively ($r_s = 0.10$, $p = 0.112$, $r_s = 0.03$, $p = 0.665$; S5A and S5B Fig). As an increase in sound level was observed at 1 h after lights out compared to 4 and 6–8 h in the second night of the MSL, the correlation between %N3 and sound level was examined focusing on the first hour after lights out. The results showed a positive correlation between %N3 and sound level in the HSL ($r_s = 0.42$, $p = 0.023$; S5C Fig) and no correlation between %N3 and sound level in the MSL ($r_s = 0.25$, $p = 0.187$; S5D Fig).

Looking at the hourly differences between HSL and MSL, HSL was higher than MSL for humidity at all times (HSL first night vs MSL first night, HSL second night vs MSL second night: all $ps$ <0.05; S4B Fig). The hourly temperature and sound level average did not differ in the HSL and MSL (all $ps$ >0.05; S4A and S4C Fig).

## Comparisons of PSG measures between the HSL and MSL

Typical examples of N2 and N3 from the EEG recorded at the HSL and MSL are shown in S6 Fig. For the EEG, the sleep stage indices (K complex, sleep spindle, and slow wave) were discriminable. EEG power spectral analysis showed that the EEG spectra measured in the two laboratories were comparable (S7 Fig). The sleep variables evaluated using PSG are shown in Fig 3. The results of linear models and nonparametric tests are shown in S3 and S4 Tables. There were no main effects of place, time, or interactions for the sleep variables except for %N3 (S3 Table). There was a main effect of place on %N3 ($F_{1, 42} = 8.62$, $p = 0.005$; S3 Table); %N3 was reduced in the MSL compared with the HSL ($t_{42} = 2.94$, $p = 0.005$, d = 0.44; Fig 3I).

The SWA and SWE data are shown in Fig 4. There were no significant differences in SWA (Fig 4A and 4B) or SWE (Fig 4C and 4D) between HSL and MSL during the NREM sleep. The results in SWA during N3 were similar (S3 Table). We found that the delta-wave frequency range in C4-M1 during NREM sleep showed a significant main effect of place ($F_{1, 42} = 4.80$, $p = 0.034$; S3 Table). The HSL delta-wave activity in C4-M1 was significantly higher than MSL ($t_{42} = 2.19$, $p = 0.034$). There were significant differences in alpha-wave activity and sigma-wave activity in C4-M1 between HSL and MSL ($F_{1, 13.7} = 6.89$, $p = 0.020$, $F_{1, 13.9} = 11.26$, $p = 0.005$, respectively; S3 Table). Alpha-wave and sigma-wave activity significantly increased in HSL compared with in MSL in C4-M1 ($t_{13.7} = 2.62$, $p = 0.020$, $t_{13.9} = 3.35$, $p = 0.005$, respectively). There was a main effect of place on beta-wave activity in F3-M2 and O1-M2 between the two laboratories ($F_{1, 42} = 5.19$, $p = 0.028$, $F_{1, 42} = 5.73$, $p = 0.021$, respectively; S3 Table). Beta-wave activity showed significant increments in HSL compared with MSL in F3-M2 and O1-M2 ($t_{42} = 2.28$, $p = 0.028$, $t_{42} = 2.39$, $p = 0.021$, respectively).

The results of the OSA-MA are shown in Fig 5. There were no significant differences in "sleepiness on rising," "frequent dreaming," "refreshness," and "sleep length" (Fig 5A, 5C–5E and S4 Table, respectively) between the HSL and MSL. A main effect of place was only found for "initiation and maintenance of sleep" ($F_{1, 42} = 7.27$, $p = 0.010$; S3 Table), with scores for the HSL significantly higher than those for the MSL ($t_{42} = 2.70$, $p = 0.010$, d = 0.84; Fig 5B). Furthermore, we found a significant correlation between %N3 and "initiation and maintenance of sleep" in OSA-MA ($r = 0.31$, $p = 0.016$).

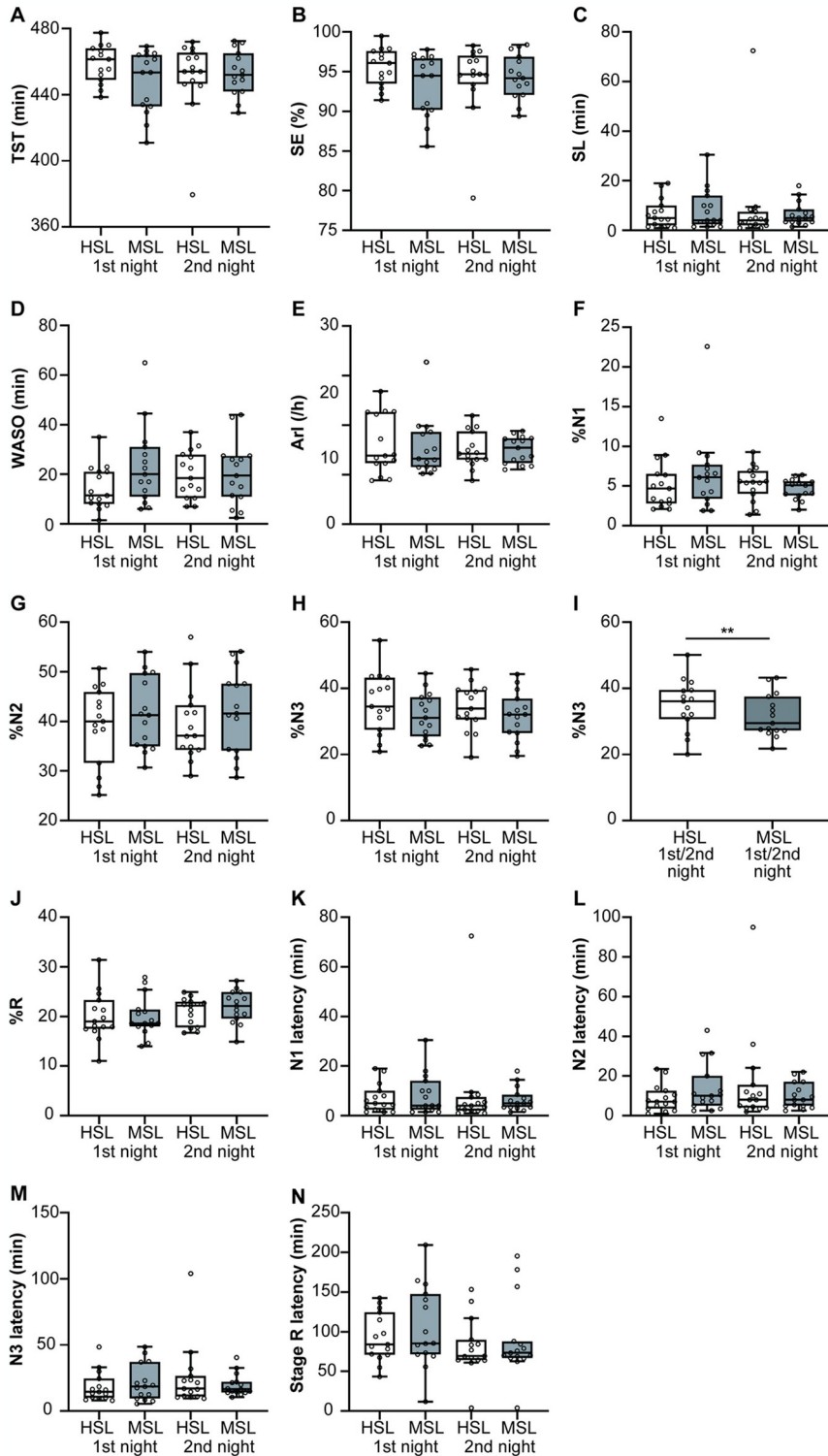

**Fig 3. Sleep variables assessed using polysomnography in the Human Sleep Lab (HSL) and Mobile Sleep Lab (MSL).** Results from the first and second nights are illustrated using Tukey-style box-and-whisker plots. White boxes represent the HSL, and gray boxes represent the MSL. (A) Total sleep time, (B) sleep efficiency, (C) sleep latency, (D) wake after sleep onset, (E) arousal index, (F) %N1, (G) %N2, (H) %N3, (I) %N3 on the first or second night in the HSL and the first or second night in the MSL, (J) %R, (K) N1 latency, (L) N2 latency, (M) N3 latency, and (N) stage R latency are shown. HSL, Human Sleep Lab; MSL, Mobile Sleep Lab; **$p < 0.01$.

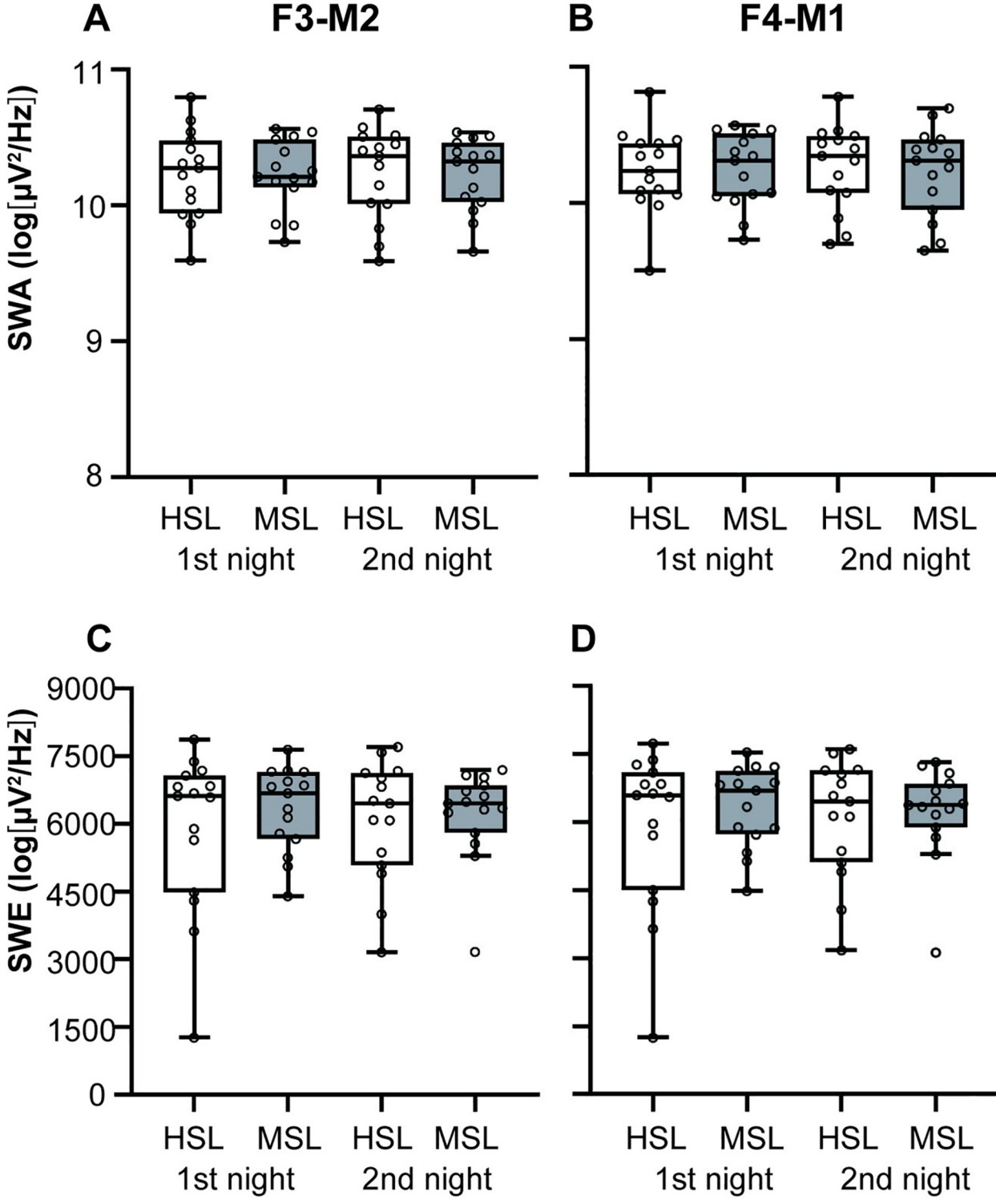

**Fig 4. Slow-wave activity (SWA) and slow-wave energy (SWE) during stages N2 and N3, illustrated using Tukey-style box-and-whisker plots.** White boxes indicate the HSL, and gray boxes indicate the MSL. SWA: (A) F3-M2, (B) F4-M2. SWE: (C) F3-M2, (D) F4-M1. SWA and SWE are natural log-transformed. HSL, Human Sleep Lab; MSL, Mobile Sleep Lab; SWA, slow-wave activity; SWE, slow-wave energy.

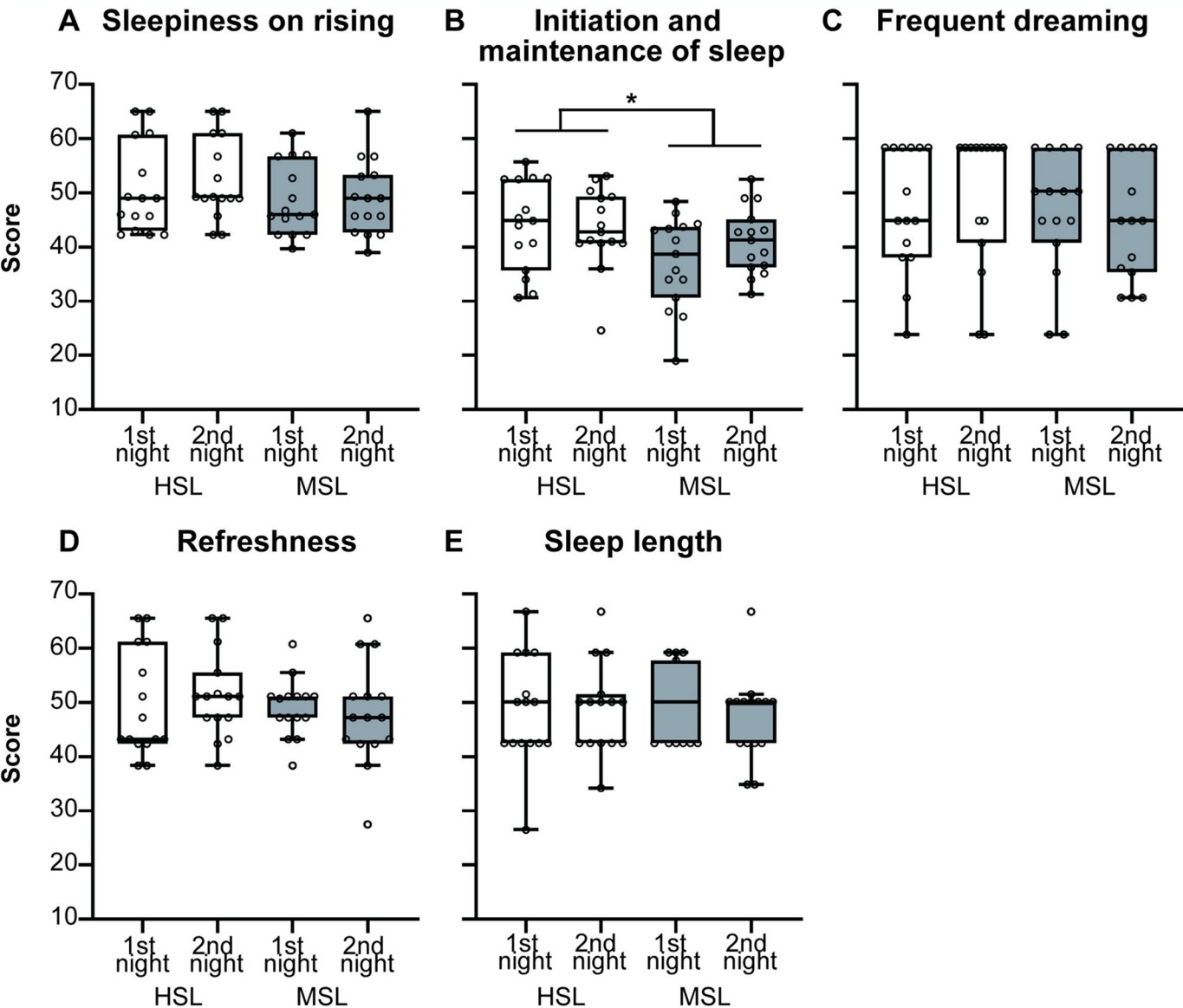

**Fig 5. Self-reported sleep quality using the Oguri-Shirakawa-Azumi Sleep Inventory, Middle-age and Aged version (OSA-MA), illustrated using Tukey-style box-and-whisker plots.** (A) Sleepiness on rising, (B) initiation and maintenance of sleep, (C) frequent dreaming, (D) refreshness, and (E) sleep length. HSL, Human Sleep Lab; MSL, Mobile Sleep Lab; *$p$ <0.05.

### Test-retest reliability of the sleep variables

ICCs were obtained using sleep variables from the HSL on the first and second nights, the MSL on the first and second nights, the HSL and MSL on the first night, the HSL and MSL on the second night, and sleep measurements (HSL or MSL) from the third and fourth nights (Fig 6). The ICCs between these two measurements showed similar reliability for the same sleep variables (Fig 6). TST, SE, WASO, %N1, and stage R latency showed poor reliability for all comparisons (Fig 6A, 6B, 6D, 6F and 6J, respectively). SL, ArI, %N2, and %R showed poor-to-moderate reliability (Fig 6C, 6E, 6G and 6I, respectively). The %N3 showed moderate-to-good reliability throughout all comparisons ($p$ <0.05, ICC = 0.57–0.79; Fig 6H).

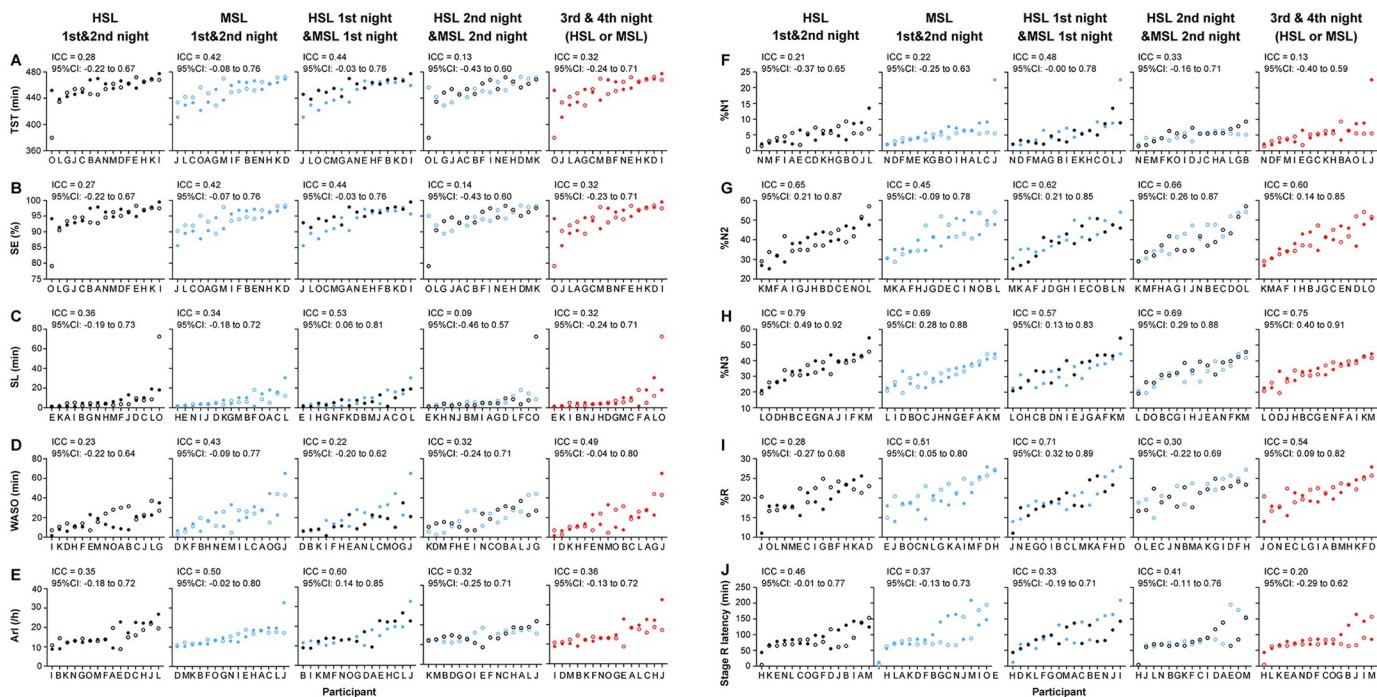

**Fig 6. Individual differences in the sleep variables assessed in the Human Sleep Lab (HSL) and Mobile Sleep Lab (MSL).** From left to right: first and second nights in the HSL, first and second nights in the MSL, first nights in the HSL and MSL, second nights in the HSL and MSL. Black-filled circles indicate the first night in the HSL, and blank black circles indicate the second night in the HSL. Blue-filled circles indicate the first night in the MSL, and blank blue circles indicate the second night in the MSL. Red-filled circles indicate the third night, and blank red circles indicate the fourth night (HSL/MSL). (A) Total sleep time, (B) sleep efficiency, (C) sleep latency, (D) wake after sleep onset, (E) arousal index, (F) %N1, (G) %N2, (H) %N3, (I) %R, and (J) stage R latency. The intraclass correlation coefficient and 95% confidence interval are shown at the top of each panel.

### Equivalence of the sleep variables

The equivalence of the acquired sleep variables was evaluated based on generic drug guidelines [35] (S8 Fig). Equivalence results are summarized in S5 Table. In all comparisons of measurements of "HSL on the first and second nights" (S8A Fig), "MSL on the first and second nights" (S8B Fig), "HSL and MSL on the first night" (S8C Fig), "HSL and MSL on the second night" (S8D Fig), and "the HSL and MSL on third and fourth nights" (S8E Fig), similar equivalence was found for the same sleep variables (S8 Fig). As noted for "sleep measurements" above, a main effect of place was found for %N3; however, evaluations showed equivalence for %N3 measured at the HSL and MSL (S8C and S8D Fig). All comparisons had equivalence for %N2, %R, ArI, and N3 latency. Furthermore, N2 latency was equivalent between HSL and MSL on the second night (S8D Fig), as it was with TST, SE, and %N1 on the first and second nights in the MSL (S8B Fig).

### Discussion

High-precision sleep testing is performed using PSG in sleep laboratories, including research facilities and hospitals. However, this may not be accessible to many individuals, including those living in remote areas. We, therefore, developed the MSL to make sleep testing universally accessible. However, the bus's environment could disturb the participant's sleep during sleep testing in the MSL. Thus, we examined whether the PSG in the MSL could perform equivalently well as that in a conventional sleep laboratory (HSL). The results revealed that it

was possible to detect characteristic EEG waveforms in each sleep stage in the MSL as well as in the HSL. We found no main effect of place on sleep variables measured in the MSL and HSL except for %N3. The %N3 showed moderate (0.5–0.75) or good (0.75–0.9) reliabilities. Although ICCs in several sleep indices were poor reliability (<0.5). Compensatory, in the %N3, equivalence was observed in the equivalence test. These results indicate that MSL was as suitable for performing PSG as the conventional sleep laboratory.

We evaluated whether the pre-experimental assessment of sleep period difference affect sleep valuables on the experimental day. During the assessment period for sleep habits, sleep duration was found to be longer on the first night than on the second night, as measured by the sleep tracker. This could be because of an order effect. A previous study reported that a group that slept 5 h had a longer TST, higher SE, shorter N1, and shorter N2 latency on the first night with 9 h of recovery sleep compared with a group that slept 9 h for one week [36]. Additionally, increased N2 and slow-wave sleep (SWS) during recovery sleep after multiple nights of sleep restriction have been reported previously [37, 38]. Here, sleep duration on the second night was shorter than that on the first night during the assessment of sleep habits; however, TST prolongation, SE increase, and other sleep variables changes that were reported in the previous studies [36–38] were not observed on the second night. Therefore, it is unlikely that sleep duration during this period affected the sleep variables during the experiment.

Regarding the environment in the bus, differences in humidity were observed between the laboratories. Although the cause could not be clarified, considering that the same number and type of dehumidifiers were placed in both rooms, the difference in humidity could be attributed to the different sizes of the rooms; the floor area of the experiment room in the HSL was 5.1 $m^2$ and that in the MSL was 3.5 $m^2$. The mean values of hourly humidity in MSLs were stable over time, and the average humidity in MSL was closer to 40–60% than that in HSL, which is considered appropriate for sleep [25]. The increased CV of MSL may be because MSL was an outdoor facility with high humidity fluctuations, and humidity was fine-tuned with a humidifier. Therefore, the MSL had a smaller room and better humidity control than the HSL.

For the sleep variables obtained from PSG, only %N3 showed a significant difference. The %N3 in the MSL was less than that in the HSL. We considered whether environmental factors were contributive to the reduction of %N3 in MSL. A 40–60% humidity is considered optimal for a sleep environment [25], and SWS reportedly decreases in humid environments [39]. Therefore, %N3 should decrease in the HSL when humidity is >60%. Based on the present contradictory results, humidity unlikely influenced the decrease in %N3 in the MSL. The sound being another environmental factor, significant fluctuation in MSL in the second night was found; the average for the first hour was higher than that in 4, 6–8 hours. Even though the hourly sound level fluctuation was relatively small, the sound level in MSL on the second night could affect the %N3 in the first part of the night. As an increased sound level might influence the decrease in %N3 in the MSL, we performed a correlation analysis. Contrary to our expectation, %N3 increased with increasing sound level during the first hour after light out in the HSL. The reason for this remains unclear. However, there was no correlation between %N3 and sound level in the MSL, so the environmental sound factor cannot explain the decrease in %N3 in the MSL.

We should discuss the possible cause for the reduction of %N3 in MSL may be stress induced by the MSL room setting. There was a difference in the ceiling height between HSL (2,110 mm [1,785 mm from bed to ceiling]) and MSL (1,627.5 mm [1,067.5 mm from bed to ceiling]). A ceiling height of 1,800–2,250 mm reportedly produced a pleasant feeling of comfortable pressure and calmness [40]; conversely, participants in spaces with low ceilings had lower positive responses (e.g., "happy," "comfortable") than those in spaces with high ceilings [41]. SWS reportedly decreases in the presence of stress [42]. The feeling of oppression caused

by the low ceilings of the MSL may have induced stress, leading to a decreased %N3. However, assuming an effect of stress due to the ceiling height, N3 latency could also be prolonged in MSL [43, 44]. To test the hypothesis that that ceiling height was perceived as stressful, assessing the stress and mood of the participants would have been necessary; however, our questionnaires in the present study did not assess this. This hypothesis could be tested in a laboratory with a manipulated room height using PSG and a refined questionnaire in a subsequent interventional study.

In the EEG frequency analysis, several frequency bands and electrodes showed MSL decrement in the EEG frequency power. Despite the %N3 reduction in MSL, there was no difference between HSL and MSL in SWA. The increase in delta waves at C4-M1 in HSL might reflect an increase in %N3; however, The AASM criteria determines N3 as frontal-induced slow wave activity (F4-M1 induced, 0.5–2 Hz, >75 μV) [18]. The decrease in MSL of C4-M1 delta activity alone cannot explain the decrease in %N3 of MSL. Therefore, frequency analysis could not fully clarify the cause of the difference in %N3 between HSL and MSL.

The frequency power values of fast waves such as alpha, sigma, and beta also increased with HSL at some electrodes during NREM, but we could not clarify the reason in the study. A tendency towards brain hyperarousal is characterized by decreased delta-power activity and increased theta-, alpha- and sigma-power activity during NREM sleep in patients with insomnia [45]. In the present study, delta-, alpha- and sigma-wave frequency power increments during NREM sleep in HSL may not indicate the brain hyperarousal tendency. At the same time, statistical analysis was performed 54 times for the frequency index. Therefore, type I errors during the multiple statistical testings might have caused these significant differences [46].

On visual inspection, no noise contamination was found in the recorded EEG (S6 Fig) in the MSL or spectral analysis (S7 Fig), and the sleep indices that characterize each sleep stage were distinguishable. Altogether, although there were several EEG frequency power decrements in the MSL, it might not entirely affect N3 determination. In the MSL, EEG could be recorded as cleanly as in the HSL and determine sleep stages.

The HSL performed better than the MSL on the self-reported sleep assessment of "initiation and maintenance of sleep." Although there were no differences between the laboratories in the PSG results of sleep latency, %N3 was higher in the HSL than in the MSL, indicating that the HSL was better in self-reported "initiation and maintenance of sleep." We found a correlation between %N3 and the "initiation and maintenance of sleep. However, since the OSA-MA combines "initiation of sleep" and "maintenance of sleep" into a single indicator, it was impossible to clarify these factors separately here. In addition, previous studies have disputed the association between %N3 and self-reported sleep depth, which has conventionally been considered an indicator of deep sleep [47]. Previously, self-reported sleep depth was most favorably associated with REM sleep but not SWS [47]. Therefore, the relationship between sleep variables and self-reported sleep requires further investigation.

The %N3 difference between the two laboratories might have been derived from the moderate to high test-retest reliability in %N3 (ICC = 0.57–0.79). Similarly, previous studies report that %N3 or N3 duration is a stable intraindividual sleep variable (%N3: ICC = 0.86, N3 duration: ICC = 0.60–0.86) [7, 48]. The high ICC of %N3 may have resulted in similar values and have been prone to a significant difference in the linear mixed model.

Except for %N3, this study showed entirely low ICC. Firstly, the sleep index's inherent variability may explain the present study's low ICC. Since SL, WASO, and stage R latency in a previous study showed medium to high night-to-night variabilities [49], these sleep indexes could result in a low ICC in the present study. Secondly, the present study design, which includes the first-night laboratory PSG, discontinuous night measure, and 2-night comparison, might cause low ICC. Thirdly, the present study's small sample size (N = 15) also contributed to the

low ICC. For a reproducible clinical study of %N3 and stage R latency, at least 20 patients need to be measured for at least three nights, and for SL and WASO, at least 50 patients and at least four nights of recording are needed [49]. By increasing the number of participants in future large-scale studies, the test-retest variability may be minimized, offering a hopeful prospect for future research.

We used a bioequivalence test to assess the equivalence of the obtained sleep variables. The bioequivalence test is a statistical method for showing that there is no difference in measurement, which is not possible in a linear mixed model or with ICC. However, while the bioequivalence test is commonly used to indicate whether a generic drug is therapeutically equivalent to the reference drug [35], it has not been applied to compare different sleep measurements. Furthermore, this study needed a larger sample size to use this method [35]. Therefore, we should consider this method as a complementary statistical method. The 90% CI for %N3 in the first- and second-night comparison between the laboratories fell within 0.0–20.0% for the different values; %N3 was more unevenly distributed in the HSL than in the MSL, although the HSL and MSL were comparable. This could explain why equivalence was observed despite the place effect. We found equivalence for similar sleep variables, including %N3, when the same participant was repeatedly measured at the same place (third and fourth nights) by comparing the measurements of the two nights in the same laboratory and between the laboratories. Although this is only based on complementary analysis, the results indicated that the MSL was as suitable for conducting sleep testing as the HSL.

FNE might have a small impact on HSL and MSL records for the following reasons. First, all sleep variables showed no main effect on timepoint (S3 Table). Second, we calculated the ICC and equivalence between the HSL on the second night and the MSL on the second night, which excluded FNE (Fig 6 and S8 Fig and S5 Table). The ICC between the HSL's second night and the MSL's second night was the same level as compared with others (Fig 6). Furthermore, N2 latency was equivalent only between the HSL's second night and the MSL's second night. However, there are few reports of the change of N2 latency in the FNE. The lack of an FNE on the sleep variables may be due to a ceiling effect caused by sleep in young healthy adults. In a future study, we should evaluate FNE across all age and patient groups, especially with insomnia.

This study has shown that MSL can measure high quality PSG in healthy adults and provide a comfortable sleep environment as HSL. The MSL has the following potential clinical applications:

1. The MSL can contribute to diagnosing narcolepsy and idiopathic hypersomnia based on the MSLT and MWT results. MSLT and MWT require examiner monitoring and real-time intervention with the participant, long test times including a pre-test PSG, and control of the environment. This study demonstrated the viability of MWT and MSLT in MSL.

2. It may be possible to adjust continuous positive airway pressure (CPAP) under manual titration in an MSL split-night study for participants not eligible for auto-CPAP [50], which may contribute to both the diagnosis and treatment of sleep apnea.

3. With an MSL, a safe PSG examination can be achieved in the presence of an examiner, and sleep disorders can be differentiated. The examiner can handle real-time artifacts such as electrode dislocation, which often occurs in children [51], and reducing artifacts improves the quality of the PSG recordings. Furthermore, in the MSL, healthcare professionals can respond to emergencies by monitoring PSG and videos, which may result in safer PSG for participants suspected of NREM parasomnias, RBD, or epilepsy. Although HST is not applicable to participants with comorbidities [16, 51], the MSL can support appropriate and

safe PSG for children and adults with comorbidities, especially older adults with potentially increased complications [52]. For example, for a participant with nocturnal hypoventilation, PSG with additional sensors such as end-tidal $CO_2$ is possible in the MSL but not the HST [51]. Thus, the MSL may help safely diagnose sleep disorders as a conventional sleep laboratory.

4. Electrode attachment by the examiner in the MSL makes it possible to accurately place the 10–20 system of EEG electrodes for diagnosing epilepsy [53] or additional EMG for diagnosing RBD [18]. Further clinical studies are needed to prove these possible benefits of MSL.

The present study has several limitations. First, many of the participants were young adults despite the absence of restrictions placed on the age of the included participants. The FNE was reportedly more significant in the laboratory than at home for older adults [13]. Because differences in location could affect sleep variables during PSG in other age groups, conducting sleep tests in participants with a wide age range is warranted to verify the equivalence of the laboratories in future studies. Second, only EEG, EMG, EOG, and ECG were measured. Sleep-related disorders, including sleep apnea, can be determined using a breathing sensor and other devices, which were not employed in this study. Future studies should confirm whether sleep-related diseases can be measured with the same accuracy in an MSL as in an HSL. Third, the sample size was small, with only 15 young volunteers. This could have led to a type II error [46]. This study did not have a sufficient sample size for bioequivalence testing [35]. While there was a significant difference between HSL and MSL in the linear mixed model for %N3, a complementary bioequivalence study showed the equivalence of %N3. However, the bioequivalence of %N3 may be due to the small sample size. Several sleep indices showed poor ICC, which may also have resulted in a small sample size for this study. More extensive studies, including studies on sleep disorders such as insomnia and sleep apnea, are necessary in a future study. Fourth, the reduction of N3 in MSL needs to be investigated with subjective and objective stress measures, using questionnaires and heart rate variability analyses, respectively. As some literature show that N3 is reduced in insomnia [54, 55], it may also be useful to validate this in a wider population that includes patients with sleep disorders, particularly insomnia. Finally, as this study did not directly compare sleep measurements using home PSG and the MSL, the added value or advantage of MSL over home sleep EEG remains unclarified. Further research is needed to clarify the advantages of the MSL over PSG at home.

## Conclusions

This study suggests that an MSL is as suitable for performing high-precision sleep testing as an HSL. The MSL can accurately examine sleep at any location. The practical application of the MSL is expected to enable sleep testing and thereby contribute to sleep medicine in remote areas, such as mountainous regions and remote islands. We anticipate that the MSL will revolutionize basic sleep research and clinical sleep medicine. In a future study, we should evaluate whether the %N3 is reduced in MSL with patients with sleep disorders, such as insomnia and sleep apnea.

## Supporting information

**S1 Fig. Consort-type diagram depicting the recruitment process of the participants.** PSG, polysomnography; BMI, body mass index.
(PDF)

**S2 Fig. Sleep time recorded using Fitbit Charge 3 during the three nights preceding the beginning of the experiment illustrated using Tukey-style box-and-whisker plots.** White boxes indicate the Human Sleep Lab (HSL), and gray boxes indicate the Mobile Sleep Lab (MSL). (A) pre-1: the day before the experiment, (B) pre-2: two nights before the experiment, (C) pre-3: three nights before the experiment, and (D) mean of the three nights. $*p < 0.05$.
(PDF)

**S3 Fig. Environmental conditions of the Human Sleep Lab (HSL) and Mobile Sleep Lab (MSL) illustrated using Tukey-style box-and-whisker plots.** White boxes indicate the HSL, and gray boxes indicate the MSL. (A) Temperature, (B) humidity, (C) sound level, and (D) vibration level. $*** p < 0.001$. No major seismic events were observed during the measurements that resulted in the D outlier, and the cause of this outlier remains unknown.
(PDF)

**S4 Fig. The average for environmental conditions during the 8-h h sleep in the Human Sleep Lab (HSL) and Mobile Sleep Lab (MSL) illustrated using Tukey-style box-and-whisker plots.** Blue boxes indicate the HSL, and red boxes indicate the MSL. (A) Temperature, (B) humidity, and (C) sound level. $*p < 0.05$.
(PDF)

**S5 Fig. Scatter plot between the average hourly sound level and %N3.** (A) The average hourly sound level and %N3 recorded in the Human Sleep Lab (HSL). (B) The average hourly sound level and %N3 recorded in the Mobile Sleep Lab (MSL). (C) The first-hour average of the sound level and %N3 recorded in the HSL. (D) The first-hour average of the sound level and %N3 recorded in the MSL.
(PDF)

**S6 Fig. Representative electroencephalographic recordings in the Human Sleep Lab (HSL) and Mobile Sleep Lab (MSL).** (A) Stage N2 and N3 recorded at the HSL. (B) Stage N2 and N3 recorded at the MSL.
(PDF)

**S7 Fig. Spectral analysis of electroencephalographic recordings in the Human Sleep Lab (HSL) and Mobile Sleep Lab (MSL).** (A) F3-M2 during the first nights in the HSL and MSL, (B) F3-M2 during the second nights in the HSL and MSL, (C) F4-M1 during the first nights in the HSL and MSL, (D) F4-M1 during the second nights in the HSL and MSL. The black line indicates the HSL, and the gray line indicates the MSL. All were natural log-transformed.
(PDF)

**S8 Fig. Equivalence of the sleep variables.** Comparisons between the first and second nights in the (A) HSL and (B) MSL; (C) between the first nights in the HSL and MSL; (D) between the second nights in the HSL and MSL; and (E) between the third and fourth nights in the HSL and MSL. Equivalence is determined when the 90% CI of the difference in the means of the sleep variables is within ±20% of the ratio of the difference. The black bars represent 90% CIs, the red-filled circles represent the mean of the difference between the means of the sleep variables, and the yellow regions show the range within ±20% of the ratio of the difference. ArI, arousal index; CI, confidence interval; HSL, Human Sleep Lab; MSL, Mobile Sleep Lab; SE, sleep efficiency; SL, sleep latency; TST, total sleep time, WASO, wake after sleep onset.
(PDF)

**S1 Table. Effect sizes of the parameters.** Effect sizes (Cohen's *d*) of the variables without an asterisk (*) were calculated using least square means and standard error from the linear mixed model. Effect sizes with an asterisk (*) were calculated using raw data. ArI, arousal index; HSL, Human Sleep Lab; MSL, Mobile Sleep Lab; SE, sleep efficiency; SL, sleep latency; TST, total sleep time; WASO, wake after sleep onset; OSA-MA, Oguri-Shirakawa-Azumi Sleep Inventory, Middle-age and Aged version.
(DOCX)

**S2 Table. Results of statistical analyses of sleep time obtained from Fitbit Charge 3 based on order, pre-exp, place, and time.** Order indicates the place (HSL or MSL) on the first and second nights and the third and fourth nights. Pre-exp represents on which night before the experiment the measurement was carried out. Place compares the HSL and MSL. Time compares the first nights (HSL and MSL) and the second nights (HSL and MSL). HSL, Human Sleep Lab; MSL, Mobile Sleep Lab.
(DOCX)

**S3 Table. Results of statistical analyses of sleep parameters, self-reported sleep quality, temperature, and coefficient of variation of environmental factors based on order, place, and time using a linear mixed model.** Order indicates the place (HSL or MSL) on the first and second nights and the third and fourth nights. Pre-exp represents on which night before the experiment the measurement was carried out. Place compares the HSL and MSL. Time compares the first nights (HSL and MSL) and the second nights (HSL and MSL). ArI, arousal index; CV, coefficient of variation; HSL, Human Sleep Lab; MSL, Mobile Sleep Lab; SE, sleep efficiency; SL, sleep latency; SWA, slow-wave activity; TST, total sleep time; WASO, wake after sleep onset; OSA-MA, Oguri-Shirakawa-Azumi Sleep Inventory, Middle-age and Aged version.
(DOCX)

**S4 Table. Results of statistical analyses of humidity, sound level, vibration level, self-reported sleep quality, and stage R latency, theta-wave activity in F3-M1 and C3-M1, and beta-wave activity in F4-M1 and C4-M1 using nonparametric tests.** OSA-MA, Oguri-Shirakawa-Azumi Sleep Inventory, Middle-age and Aged version.
(DOCX)

**S5 Table. Equivalence of sleep variables assessed in the Human Sleep Lab (HSL) and Mobile Sleep Lab (MSL).** The values for S8 Fig, of the mean of the difference between the means of the two sleep variables (90% confidence intervals).
(DOCX)

**S1 File. Dataset.**
(XLSX)

**S2 File. Terminology for each item in the S1 File.**
(DOCX)

**S3 File. Time information about light out and light on.**
(XLSX)

## Acknowledgments

The authors thank Editage (https://www.editage.com) for English language editing. The authors thank Yurina Suzuki and Yumeka Oiwa for helping data organization.

## Author Contributions

**Conceptualization:** Takashi Abe, Toshio Kokubo, Isamu Takahara, Masashi Yanagisawa.

**Data curation:** Takashi Abe.

**Formal analysis:** Chihiro Suzuki, Yoko Suzuki, Takashi Abe.

**Funding acquisition:** Masashi Yanagisawa.

**Investigation:** Chihiro Suzuki, Yoko Suzuki, Takashi Abe.

**Methodology:** Chihiro Suzuki, Yoko Suzuki, Takashi Abe.

**Project administration:** Shoji Fukusumi, Toshio Kokubo.

**Resources:** Isamu Takahara.

**Software:** Takashi Abe.

**Supervision:** Takashi Abe, Masashi Yanagisawa.

**Validation:** Chihiro Suzuki, Yoko Suzuki, Takashi Abe.

**Visualization:** Chihiro Suzuki, Yoko Suzuki, Takashi Abe.

**Writing – original draft:** Chihiro Suzuki, Yoko Suzuki, Takashi Abe.

**Writing – review & editing:** Takashi Abe, Takashi Kanbayashi, Shoji Fukusumi, Toshio Kokubo, Isamu Takahara, Masashi Yanagisawa.

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
