## [Decision Letter · Decision Letter 0]

19 Jan 2024

PONE-D-23-34836Mobile Sleep Lab: Evaluation of equivalence with a conventional sleep laboratoryPLOS ONE

Dear Dr. Suzuki,

Thank you for submitting your manuscript to PLOS ONE. After careful consideration, we feel that it has merit but does not fully meet PLOS ONE’s publication criteria as it currently stands. Therefore, we invite you to submit a revised version of the manuscript that addresses the points raised during the review process.

 In addition to addressing all of the comments made in the thorough review by the reviewer, I also have a few additional comments / questions. These are the following:  Data availability statement says that all data are available without restriction. The supplemental files contain processed data used in the reported analyses. But are raw sleep data publicly available? If yes, please add this information (and where readers can access). If not, please provide an appropriate justification. Raw data would allow interested researchers to re-score and/or conduct novel analyses not contained in this manuscript.Line 32; text states “The agreement rate between PSG and portable EEG has reached 83.5%”. Please clarify agreement rate based on what? Sleep staging? The reviewer brought up first night effects, which are known to be quite prominent. Do results hold when the first night (of each condition) is discarded? Lines 91-91; what is meant by accommodable in both the HSL and MSL Exclusion criteria on lines 101-105; please clarify whether any participants (and how many) were excluded for each of these various criteria. Based on participants section and information presented in Table 1, it looks like no participants were excluded. Please confirm.Line 115-116; one participant withdrew; only 15 included in analyses. Thus, there could not be 8 in each group with respect to order. Please clarify. Please submit your revised manuscript by Mar 04 2024 11:59PM. If you will need more time than this to complete your revisions, please reply to this message or contact the journal office at plosone@plos.org. Please include the following items when submitting your revised manuscript:A rebuttal letter that responds to each point raised by the academic editor and reviewer(s). You should upload this letter as a separate file labeled 'Response to Reviewers'.A marked-up copy of your manuscript that highlights changes made to the original version. You should upload this as a separate file labeled 'Revised Manuscript with Track Changes'.An unmarked version of your revised paper without tracked changes. You should upload this as a separate file labeled 'Manuscript'.

We look forward to receiving your revised manuscript.

Kind regards,

Bradley R. King

Academic Editor

PLOS ONE

“This work was supported by the “Social Application of Mobility Innovation and Future Social Engineering Research Phase IV,” a joint research project between Toyota Motor Corporation and the University of Tsukuba; and by the Ministry of Education, Culture, Sports, Science and Technology (MEXT), World Premier International Research Center Initiative (WPI) program; and Japan Agency for Medical Research and Development (AMED) under Grant Number JP21zf0127005. Dr. Masashi Yanagisawa's work has been funded by the “Social Application of Mobility Innovation and Future Social Engineering Research Phase IV,” a joint research project between Toyota Motor Corporation and the University of Tsukuba; and by the MEXT, WPI program; and AMED under Grant Number JP21zf0127005.”

“I have read the journal's policy and the authors of this manuscript have the following competing interests: The hydrogen-fueled bus used in the study was rented free of charge from Toyota Motor Cooperation. Dr. Yanagisawa's work has been funded by the “Social Application of Mobility Innovation and Future Social Engineering Research Phase IV,” a joint research project between Toyota Motor Corporation and the University of Tsukuba; and by the MEXT, WPI program; and AMED under Grant Number JP21zf0127005. Dr. Takahara was a former employee of Toyota Motor Cooperation, and the general manager in charge of the Future Development Office, Frontier Research Center, Toyota Motor Corporation. The other authors declare no potential conflict of interest.”

5. We note that Figure 1 in your submission contain copyrighted images. All PLOS content is published under the Creative Commons Attribution License (CC BY 4.0), which means that the manuscript, images, and Supporting Information files will be freely available online, and any third party is permitted to access, download, copy, distribute, and use these materials in any way, even commercially, with proper attribution. For more information, see our copyright guidelines: http://journals.plos.org/plosone/s/licenses-and-copyright.

6. We are unable to open your Supporting Information file [S1_Fig, S2_Fig, S3_Fig, S4_Fig, S5_Fig]. Please kindly revise as necessary and re-upload.

Reviewers' comments:

Reviewer's Responses to Questions

**Comments to the Author**

1. Is the manuscript technically sound, and do the data support the conclusions?

Reviewer #1: Yes

2. Has the statistical analysis been performed appropriately and rigorously? 

Reviewer #1: Yes

3. Have the authors made all data underlying the findings in their manuscript fully available?

Reviewer #1: Yes

4. Is the manuscript presented in an intelligible fashion and written in standard English?

Reviewer #1: Yes

5. Review Comments to the Author

Reviewer #1: The purpose of this study was to compare sleep quality between a mobile sleep lab housed in a bus and a conventional sleep laboratory. Fifteen young healthy adults participated in the study, sleeping for 4 nights with polysomnography (PSG): 2 nights in the mobile lab and 2 nights in the conventional lab, with the order counterbalanced. The findings revealed a decrease in the time spent in N3 sleep stage in the mobile sleep lab. Equivalence for the different labs and nights was also computed and revealed that sleep variables were equivalent across labs and nights. Authors concluded that the mobile sleep lab was equivalent to the conventional Sleep Lab regarding suitability for conducting PSG.

A notable strength of this study is that the mobile lab offers direct access to healthcare and sleep expertise for individuals residing in remote areas without access to hospitals or sleep units. This ambulatory technology has the potential to facilitate the diagnosis of sleep disorders, as well as the treatment and follow-up of patients to assess treatment effectiveness. However, it is crucial to acknowledge that the study focused on young individuals with good sleep patterns. Therefore, further research is needed to evaluate the performance of this mobile sleep lab in diverse patient populations, particularly those with insomnia.

I have comments hereafter that the authors should address.

- The main interest to develop this mobile sleep lab is to assess patients with sleep disorders. However, only young and good sleepers were assessed in this study. It is striking to see that deep N3 sleep stage is reduced in the mobile lab in healthy young subjects. Further research is needed to evaluate the performance of this mobile sleep lab in diverse patient populations, particularly those with insomnia. This is important, especially if this decrease in N3 is due to a higher level of stress as suggested in the discussion section (even if speculative). I suggest highlighting it in the conclusions and the abstract of the paper.

- Another important question raised by this study is: what is the added value of this mobile sleep lab to home sleep EEG? Authors addressed this point in the discussion section. However, they also stated (Page 3, lines 34-36): “Although current portable EEG devices can accurately determine sleep stages, it is difficult to simultaneously measure respiratory variables and perform electromyography (EMG) of the lower limbs, which are crucial for investigating sleep disorders such as sleep apnea”. I am concerned that this assertion may not be accurate, as several studies have utilized ambulatory EEG in home settings, demonstrating its efficiency, for instance, in investigating sleep disorders like OSA (e.g., Bruyneel et al., 2011). Additionally, individuals sleeping in their familiar environment may experience improved sleep quality compared to a night of sleep in the lab (e.g., Iber et al, 2004). It has been shown that the first-night effect, typically observed in unfamiliar environments like hospitals, is significantly reduced in PSG at home (e.g., Edinger et al., 1997). I recommend rephrasing the mentioned sentence in this context, emphasizing the advantages of PSG at home, and not solely focusing on the drawbacks to provide a more comprehensive perspective. Furthermore, it is crucial to discuss the limitation of not comparing sleep at home directly, which warrants further exploration.

- We lack information about the EEG settings. What kind of EEG system was used for PSG? How many electrodes? Which locations? Was a 10-20 EEG montage used? This information should be mentioned precisely in the method section.

- How was the arousal index calculated?

- EEG power spectral analyses:

o SWA was averaged for N2 and N3 sleep stages. I suggest to also compute power spectral analysis on N3 only.

o SWA was averaged across right frontal, central and occipital electrodes. However, slow oscillations are prominent in frontal locations. I would recommend computing the analyses on frontal derivations. Moreover, the analyses are computed on electrodes located on the right side of the head. Why?

o Regarding SWS, distinction between slow oscillations (SO) frequency range (0.5–1 Hz) and delta frequency range (1–4 Hz) has been done in the literature (e.g., Mander et al., 2015; Winer et al., 2019; Kim et al., 2019). I suggest to also analyze SO and delta waves separately.

o Finally, I also recommend running spectral analyses on other frequency ranges (i.e., theta = 4–8 Hz, alpha = 8–12 Hz, sigma = 12–16 Hz, and beta = 16–30 Hz).

- A bioequivalence method is used in this article to compare sleep variables between the 2 labs and the different nights. This method is commonly used to compare 2 drugs, typically to indicate whether a new drug is therapeutically equivalent to the reference drug. Although I am not familiar with the bioequivalence method, I have never seen it applied to compare different sleep measurements. The rationale to use this method in this context should be better stated as well as its complementarity with the different statistical methods used in this article (i.e., linear mixed models and intraclass correlation). Moreover, the authors should also interpret potential discrepancies in the results obtained using these different methods.

- ICCs indicate poor reliability for several sleep variables. How to explain that?

- Page 16 (line 46-147). Authors wrote: “… from the afternoon of each day, they were asked to abstain from smoking….” Did smokers were included? If yes, abstinence can affect their subsequent night of sleep, and this should be discussed. If not, this sentence is misleading and should be modified.

- The sample size is quite small with only 15 young volunteers included. It should be discussed in the limitation section.

- N3 decrease interpretation.

o Authors speculates that N3 decrease might be due to an increase level of stress due to a lower ceiling in the mobile lab. However, they did not question the participants on their stress level nor on how they feel in the lab. Moreover, I would expect elevated levels of stress to also delay N3 sleep latency, which is not the case. I recommend acknowledging the absence of questionnaires assessing participants' feelings to validate the authors' hypothesis regarding higher stress level in the mobile lab.

o To determine whether there is a link between the lower % N3 and the lower self-reported “initiation and maintenance of sleep” in the mobile sleep lab, I suggest computing correlational analyses.

- The authors extensively discussed the potential causes for the difference in the level of humidity in the two labs. Based on Fig. S3, it seems that the averaged level of humidity is in the normal range (around 50-60%) or a bit higher in the conventional lab. Does the level of humidity fluctuate throughout the night? If yes, does this fluctuation differ between the 2 labs? (same question for temperature and noise). For noise, if it is higher during the beginning of the night (due for example to lower external soundproofing in the bus), it can affect the first part of the night which is richer in SWA.

6. PLOS authors have the option to publish the peer review history of their article (what does this mean?). If published, this will include your full peer review and any attached files.

Reviewer #1: No

---

## [Author Response · Author response to Decision Letter 0]

18 Sep 2024

Response to Reviewers

Reviewers' comments:

Editor #1_1: Data availability statement says that all data are available without restriction. The supplemental files contain processed data used in the reported analyses. But are raw sleep data publicly available? If yes, please add this information (and where readers can access). If not, please provide an appropriate justification. Raw data would allow interested researchers to re-score and/or conduct novel analyses not contained in this manuscript.

Response: Thank you for your comment. We agree with your suggestion, and we have added raw sleep polysomnographic data as Supporting Information (S3 File). Polysomnographic data (S3 File) is available in Zenodo at https://doi.org/10.5281/zenodo.12742345.

The private access URL is here: 

https://zenodo.org/uploads/12742345?token=eyJhbGciOiJIUzUxMiIsImlhdCI6MTcyMTMwNDM2NywiZXhwIjoxNzUzOTE5OTk5fQ.eyJpZCI6IjgyOTMyN2M3LWQyMmItNDI0OC05NjY0LWM1NDU4YzM0OTczMiIsImRhdGEiOnt9LCJyYW5kb20iOiIzMDZlOGVlMmFjOWY3ZjA0ZDAxYzcwMWI1ZjY1NDRlYiJ9.cvZDzZahxyyr3220mPSF60K63NrHKhaDsEe-UvtMK-QLTU33aIlZP8fXZD4jN-lakJUB3NY6f45b_TcPYl_Cjg. 

We have added the following explanation for the PSG data on page 10, lines 209-211: “The output from the Polymate Pro was in the European Data Format (EDF), setting the high-pass filter to 0.03 Hz and the low-pass filter to 200 Hz (S3 File).”

Editor #1_2: Line 32; text states “The agreement rate between PSG and portable EEG has reached 83.5%”. Please clarify agreement rate based on what? Sleep staging?

Response: Thank you for your comments and questions. The agreement rate was calculated based on sleep stage scoring. We have changed the text, on page 3, lines 35-37: “The agreement rate based on sleep stage scoring between PSG and portable EEG has reached as high as 83.5% [3]; thus, portable EEG can be used to accurately measure sleep EEG.”

Editor #1_3: The reviewer brought up first night effects, which are known to be quite prominent. Do results hold when the first night (of each condition) is discarded? 

Response: Thank you for this pertinent question. Yes, these do. As the ICC excludes the first night, we calculated the ICC and equivalence between the HSL’s second and the MSL’s second night. The ICCs between the HSL’s second and MSL’s second night were the same level as others. The equivalence of N2 latency was shown only between HSL’s second night and MSL’s second night. However, there are few reports of changes in N2 latency in the first-night effects. We have added the following sentences from page 24, line 573 to page 25 line 582: “FNE might have a small impact on HSL and MSL records for the following reasons. First, all sleep variables showed no main effect on timepoint (S3 Table). Second, we calculated the ICC and equivalence between the HSL on the second night and the MSL on the second night, which excluded FNE (Fig 6, S7 Fig, S5 Table). The ICC between the HSL’s second night and the MSL’s second night was the same level as compared with others (Fig 6). Furthermore, N2 latency was equivalent only between the HSL’s second night and the MSL’s second night. However, there are few reports of the change of N2 latency in the FNE. The lack of an FNE on the sleep variables may be due to a ceiling effect caused by sleep in young healthy adults. In a future study, we should evaluate FNE across all age and patient groups, especially with insomnia.”

Editor #1_4: Lines 91-91; what is meant by accommodable in both the HSL and MSL

Response: Thank you for this question. We meant that people who can stay and sleep overnight in both HSL and MSL. We have modified on page 6, lines 104-105: “those who can stay and sleep overnight”

Editor #1_5: Exclusion criteria on lines 101-105; please clarify whether any participants (and how many) were excluded for each of these various criteria. Based on participants section and information presented in Table 1, it looks like no participants were excluded. Please confirm. 

Response: Thank you for bringing this to our attention. We recruited 25 participants, 4 participants withdrew consent, one had a history of PSG and four participants were excluded because their BMI was ineligible. 1 participant withdrew consent after the first PSG measurement (S1 Fig). The characteristics of the final 15 participants are summarized in Table 1. We have added the following sentences on page 6, lines 118-120: ”We recruited 25 participants; four withdrew consent, one had a history of PSG and another four were excluded due to their ineligible BMI (S1 Fig).”

Editor #1_6: Line 115-116; one participant withdrew; only 15 included in analyses. Thus, there could not be 8 in each group with respect to order. Please clarify.

Response: Thank you for your suggestion. seven participants underwent PSG in the order of HSL→MSL, on the other hand, eight underwent PSG in the order of MSL→HSL. We have added the following sentences on page 7, lines 137-139: “Since one participant withdrew, seven underwent PSG first at the HSL, followed by the MSL. In contrast, eight underwent PSG in the order of MSL, followed by HSL.”

Journal requirement_1: 1. Please ensure that your manuscript meets PLOS ONE's style requirements, including those for file naming. The PLOS ONE style templates can be found at

Response: Thank you for your comment and information. We ensured that the manuscript met the criteria of PLOS ONE.

Journal requirement_2: Note from Emily Chenette, Editor in Chief of PLOS ONE, and Iain Hrynaszkiewicz, Director of Open Research Solutions at PLOS: Did you know that depositing data in a repository is associated with up to a 25% citation advantage (https://doi.org/10.1371/journal.pone.0230416)? If you’ve not already done so, consider depositing your raw data in a repository to ensure your work is read, appreciated and cited by the largest possible audience. You’ll also earn an Accessible Data icon on your published paper if you deposit your data in any participating repository (https://plos.org/open-science/open-data/#accessible-data).

Response: Thank you for this suggestion and information. We have uploaded the raw data of the polysomnogram data (S3 File) in a repository. 

Journal requirement_3: Thank you for stating the following financial disclosure:

“This work was supported by the “Social Application of Mobility Innovation and Future Social Engineering Research Phase IV,” a joint research project between Toyota Motor Corporation and the University of Tsukuba; and by the Ministry of Education, Culture, Sports, Science and Technology (MEXT), World Premier International Research Center Initiative (WPI) program; and Japan Agency for Medical Research and Development (AMED) under Grant Number JP21zf0127005. Dr. Masashi Yanagisawa's work has been funded by the “Social Application of Mobility Innovation and Future Social Engineering Research Phase IV,” a joint research project between Toyota Motor Corporation and the University of Tsukuba; and by the MEXT, WPI program; and AMED under Grant Number JP21zf0127005.”

Response: Thank you for bringing this to our attention. We have added the following sentences in the cover letter, “The funders had no role in study design, data collection and analysis, decision to publish, or preparation of the manuscript.”

Journal requirement_4: Thank you for stating the following in the Competing Interests section:

“I have read the journal's policy and the authors of this manuscript have the following competing interests: The hydrogen-fueled bus used in the study was rented free of charge from Toyota Motor Cooperation. Dr. Yanagisawa's work has been funded by the “Social Application of Mobility Innovation and Future Social Engineering Research Phase IV,” a joint research project between Toyota Motor Corporation and the University of Tsukuba; and by the MEXT, WPI program; and AMED under Grant Number JP21zf0127005. Dr. Takahara was a former employee of Toyota Motor Cooperation, and the general manager in charge of the Future Development Office, Frontier Research Center, Toyota Motor Corporation. The other authors declare no potential conflict of interest.”

Response: Thank you for this pertinent comment. We confirmed that this does not alter our adherence to all PLOS ONE policies on sharing data and materials. We have no restrictions on sharing data and materials. We have added, “This does not alter our adherence to PLOS ONE policies on sharing data and materials.” We have added the latest Competing Interests statement in the cover letter.

Journal requirement_5: We note that Figure 1 in your submission contain copyrighted images. All PLOS content is published under the Creative Commons Attribution License (CC BY 4.0), which means that the manuscript, images, and Supporting Information files will be freely available online, and any third party is permitted to access, download, copy, distribute, and use these materials in any way, even commercially, with proper attribution. For more information, see our copyright guidelines: http://journals.plos.org/plosone/s/licenses-and-copyright.

Response: Thank you for your comments. We understand permission is needed to use the photograph of the International Institute for Integrative Sleep Medicine (WPI-IIIS) logo and exterior design of the Mobile Sleep Lab in Figure 1A. We have uploaded written permission from the copyright holder of the WPI-IIIS logo (content-permission-form_1) and the designer, Tadanobu Hara, who designed the exterior of the Mobile Sleep Lab (content-permission-form_2), to publish these figures under the CC BY 4.0 license. 

 We have added to the figure caption for Figure 1 on page 5, lines 80-83: “The logo of the International Institute for Integrative Sleep Medicine (WPI-IIIS) in this Fig 1A is used with permission from the institute. The photograph (Fig 1A), including the exterior design of the Mobile Sleep Lab has been used with permission from the designer, Tadanobu Hara.”

 We contacted the Frontier Research Center of Toyota Motor Corporation for permission to use the photograph of the fuel cell bus. Toyota Motor Corporation has already published the following information on its website, and the view was that academic publications would be no problem as long as the citation or source was stated. We have obtained written permission from the Frontier Research Center of Toyota Motor Corporation (content-permission-form_3).

We have added the citation number and the following sentence in the Figure 1 caption on page 5, lines 83-84: “Permission to use the photographs and images (Fig 1A-C) of the fuel cell bus was obtained from Toyota Motor Corporation [17].”

We have added the reference as follows on page 30, lines 710-714: Toyota Motor Corporation. [Japanese]; 2022 November 18 [cited 2024 August 2]. Available from: https://global.toyota/jp/mobility/frontier-research/38233999.html?_gl=1*1gnk8f2*_ga*MTEzOTczOTQ4OS4xNjQ3NDA4Njgx*_ga_FW87SM9FNZ*MTcyMjUwMjI2Ni4xMi4xLjE3MjI1MDIzODQuMzAuMC4w&_ga=2.153087097.118242171.1722502267-1139739489.1647408681

Journal requirement_6: We are unable to open your Supporting Information file [S1_Fig, S2_Fig, S3_Fig, S4_Fig, S5_Fig]. Please kindly revise as necessary and re-upload.

Response: Thank you for your confirmation. We are sorry for our inattentiveness. We have revised and re-uploaded the Supporting Information.

Reviewer #1_1: Is the manuscript technically sound, and do the data support the conclusions?

Reviewer #1: Yes.

Response: Thank you for your confirmation. 

Reviewer #1_2: Has the statistical analysis been performed appropriately and rigorously?

Reviewer #1: Yes

Response: Thank you for your confirmation.

Reviewer #1_3: Have the authors made all data underlying the findings in their manuscript fully available?

Reviewer #1: Yes

Response: Thank you for your confirmation. 

Reviewer #1_4: Is the manuscript presented in an intelligible fashion and written in standard English?

PLOS ONE does not copyedit accepted manuscripts, so the language in submitted articles must be clear, correct, and unambiguous. Any typographical or grammatical errors should be corrected at revision, so please

---

## [Decision Letter · Decision Letter 1]

25 Oct 2024

PONE-D-23-34836R1Mobile Sleep Lab: Comparison of polysomnographic parameters with a conventional sleep laboratoryPLOS ONE

Dear Dr. Suzuki,

Thank you for submitting your manuscript to PLOS ONE. We appreciate your efforts to update the manuscript and provide detailed and thoughtful responses. After careful consideration, we feel that it has merit but does not fully meet PLOS ONE’s publication criteria as it currently stands. Therefore, we invite you to submit a revised version of the manuscript that addresses the remaining points. 

 The reviewer listed their remaining comments below. On my end, the link to Zenodo did not work (I did not have the appropriate privileges for the private link). I encourage you to make it public or ensure we can check the availability of the data. Please also add a statement in the main text that the processed data are available as supplemental information and the raw data are provided without restriction on zenodo (with the correct link). This will facilitate data re-use. 

We look forward to receiving your revised manuscript.

Kind regards,

Bradley R. King

Academic Editor

PLOS ONE

Journal Requirements:

Reviewers' comments:

Reviewer's Responses to Questions

**Comments to the Author**

1. If the authors have adequately addressed your comments raised in a previous round of review and you feel that this manuscript is now acceptable for publication, you may indicate that here to bypass the “Comments to the Author” section, enter your conflict of interest statement in the “Confidential to Editor” section, and submit your "Accept" recommendation.

Reviewer #1: (No Response)

2. Is the manuscript technically sound, and do the data support the conclusions?

Reviewer #1: Yes

3. Has the statistical analysis been performed appropriately and rigorously? 

Reviewer #1: Yes

4. Have the authors made all data underlying the findings in their manuscript fully available?

Reviewer #1: Yes

5. Is the manuscript presented in an intelligible fashion and written in standard English?

Reviewer #1: Yes

6. Review Comments to the Author

Reviewer #1: Thank you for the thorough revisions you have made to the manuscript. I have a few remaining comments regarding the statistical analyses that need to be addressed:

- On page 22 (lines 511-520), the authors mentioned that spectral power analyses were performed without corrections for multiple comparisons. I recommend applying appropriate corrections for multiple comparisons across all the analyses and reporting results that survive these corrections.

- Similarly, regarding the new analyses conducted on sound level, there was an effect of time during the 8-hour period in the mobile sleep lab (MSL), with higher sound level observed in the first hour compared to later hours in the second half of the night. Were these analyses also corrected for multiple comparisons? Please clarify this.

- Additionally, authors reported a correlation between %N3 and subjective sleep initiation and maintenance. Did they test whether %N3 also correlates with sound level? Given that the authors hypothesize that the higher sound level in the first hour may explain the reduced %N3 in the MSL, it would be useful to examine this relationship. If a correlation exists between sound level and %N3, it could further support the interpretation that the reduced %N3 is attributable to environmental factors (e.g., sound) rather than stress. This would strengthen the conclusion that sleep parameters in the MSL are comparable to those obtained with conventional polysomnography, if sound levels are adequately controlled throughout the night.

Finally, there is a minor correction needed in Figure 2: the number of participants is noted as N = 16, but the final sample includes 15 participants. Please adjust or remove this notation accordingly.

7. PLOS authors have the option to publish the peer review history of their article (what does this mean?). If published, this will include your full peer review and any attached files.

Reviewer #1: No

---

## [Author Response · Author response to Decision Letter 1]

14 Nov 2024

Response to Reviewers

Reviewers' comments:

Editor #1_1: On my end, the link to Zenodo did not work (I did not have the appropriate privileges for the private link). I encourage you to make it public or ensure we can check the availability of the data. Please also add a statement in the main text that the processed data are available as supplemental information and the raw data are provided without restriction on zenodo (with the correct link). This will facilitate data re-use.

Response: Thank you for your comment. We are sorry for the inappropriate link, and we have published the data on Zenodo and modified the link. 

We have added the following sentence on page 10, lines 211-213: “The processed data are available as supplemental information and the raw data are provided without restriction on Zenodo (https://doi.org/10.5281/zenodo.14020530) [27].” We have also added the following sentence to the Supporting Information, page 38, lines 932-933: “The raw data are available without restriction on Zenodo (https://doi.org/10.5281/zenodo.14020530) [27].” We have added reference number 27 for the published data: “Suzuki C, Suzuki Y, Abe T, Kanbayashi T, Fukusumi S, Kokubo T et al. Mobile Sleep Lab: Comparison of polysomnographic parameters with a conventional sleep laboratory [Data set]. Zenodo. 2024; https://doi.org/10.5281/zenodo.14020530.” 

　We also found an error in one record of the sleep staging data used for frequency analysis. The S3 File and the frequency analysis results in the manuscript, S1 File (highlighted in yellow), and S3 Table (highlighted in blue) have been modified. The non-parametric tests results of the frequency analysis and the summary of the results and discussion are unchanged.

Journal requirement_1: Please review your reference list to ensure that it is complete and correct. If you have cited papers that have been retracted, please include the rationale for doing so in the manuscript text, or remove these references and replace them with relevant current references. Any changes to the reference list should be mentioned in the rebuttal letter that accompanies your revised manuscript. If you need to cite a retracted article, indicate the article’s retracted status in the References list and also include a citation and full reference for the retraction notice.

Response: Thank you for your comment and information. We rechecked the reference list to see whether it is complete and correct. We cited the first guideline for the 10-20 system in the previous manuscript. However, since there has been a subsequent revision of the guideline, we have replaced the cited references with the revised version. We replaced reference 26 as follows: Klem GH, Lüders HO, Jasper HH, Elger C. The ten-twenty electrode system of the International Federation. The International Federation of Clinical Neurophysiology. Electroencephalogr Clin Neurophysiol Suppl. 1999;52: 3-6.

Reviewer #1_1: Thank you for the thorough revisions you have made to the manuscript. I have a few remaining comments regarding the statistical analyses that need to be addressed:

- On page 22 (lines 511-520), the authors mentioned that spectral power analyses were performed without corrections for multiple comparisons. I recommend applying appropriate corrections for multiple comparisons across all the analyses and reporting results that survive these corrections.

Response: Thank you for your confirmation. We apologize – the statement that no correction is made was incorrect, since Bonferonni correction is made at each electrode. Since the p-value results in the manuscript are the results after Bonferroni correction, the results remain the same. We did not correct the inter-electrode p-values because we did not evaluate the topographic distribution of the power values in the frequency analysis. 

The text was modified as follows on page 23, lines 544-545: “At the same time, statistical analysis was performed 54 times for the frequency index.” by deleting the phrase “and each electrode with no correction”

Reviewer #1_2: Similarly, regarding the new analyses conducted on sound level, there was an effect of time during the 8-hour period in the mobile sleep lab (MSL), with higher sound level observed in the first hour compared to later hours in the second half of the night. Were these analyses also corrected for multiple comparisons? Please clarify this.

Response: Thank you for your confirmation. Bonferroni correction for multiple comparisons was already used in the sound level analysis in the previous manuscript. We did not describe this fact, so we added the sentence in the Methods on page 15, lines 321-322: “For multiple comparisons, Bonferroni correction was used.”

In relation to this comment, we have reviewed the analysis and have changed our response for the previous reviewer #1_5_11's comment: The authors extensively discussed the potential causes for the difference in the level of humidity in the two labs. Based on Fig. S3, it seems that the averaged level of humidity is in the normal range (around 50-60%) or a bit higher in the conventional lab. Does the level of humidity fluctuate throughout the night? If yes, does this fluctuation differ between the 2 labs? (same question for temperature and noise). For noise, if it is higher during the beginning of the night (due for example to lower external soundproofing in the bus), it can affect the first part of the night which is richer in SWA. In the previous manuscript, we evaluated temperature, humidity, and sound level fluctuations overnight by calculating the hourly coefficient of variations (CVs), but we thought that calculating CVs over 8 h and quantitatively evaluating them would be a more appropriate way to compare between the HSL and MSL, rather than comparing hourly CVs. Therefore, we recalculated the CVs from 8-h recordings. CV results and explanations were added to the S1 File (highlighted in green), S3 Table (highlighted in yellow), and S2 File (highlighted in yellow), respectively. The results showed that the CVs for temperature and humidity were higher in the MSL, while noise fluctuations were lower in the MSL compared to the HSL. Related to this change, pannels in the S4 Figure for each hourly CV evaluation were removed. We have modified the text in the Methods from page 9, line 191 to page 10, line 194: “We analyzed the temperature, humidity, sound, vibration, and the coefficient of variation (CV) of each parameter only during the 8 h of nighttime sleep. We calculated the average temperature, humidity, and sound in each hour separately, and the CV in each hour separately.”. We have added following the results at page 15, line 343 to page 16, line 349: “There were main effects of place in CVs on temperature, humidity, and sound between the two laboratories (F1, 14 = 6.96, p = 0.020, F1, 42 = 51.20, p <0.0001, F1, 42 = 14.05, p <0.001, respectively; S3 Table). CVs of temperature and humidity showed significant increases in the MSL compared with the HSL (t14 = −2.64, p = 0.020, t42 = −7.16, p <0.0001, respectively). On the contrary, the CV of sound level significantly decreased in the MSL compared to the HSL (t42 = 3.75, p <0.001).” The discussion is unchanged.

We apologize for our mistake; in the previous manuscript, the statistical results in the hourly evaluation of environmental factors in the manscript were only sound results. Furthermore, the statistical results in the S4 Figure were also insufficient and difficult to understand. We have added hourly averaged temperature and humidity results in the manuscript, and all statistical results for temperature, humidity, and sound levels are now shown in S4 Figure using asterisks. We have added the following results on page 16, lines 350-358: “The average was calculated every hour to examine the effects of changes in environmental measurements on sleep over time (S4 Fig). In temperature, a main effect of time was found for MSL (first night: F(7) = 25.03, p <0.001; second night: F(7) = 17.5, p = 0.014). No significant differences by time for both MSL nights were found in post-hoc tests (all ps >0.05, S4 Fig. A). In humidity, a main effect of time was found for HSL (first night: F(7) = 45.4, p <0.0001; second night: F(7) = 22.1, p = 0.002). Post-hoc tests revealed a significant increase in humidity at hour 7 versus hour 3 for the first night of HSL (Z = −3.351, p = 0.023, S4 Fig. B). There were no significant differences in post-hoc tests on the second night at the HSL (all ps >0.05).”.

Reviewer #1_3: Additionally, authors reported a correlation between %N3 and subjective sleep initiation and maintenance. Did they test whether %N3 also correlates with sound level? Given that the authors hypothesize that the higher sound level in the first hour may explain the reduced %N3 in the MSL, it would be useful to examine this relationship. If a correlation exists between sound level and %N3, it could further support the interpretation that the reduced %N3 is attributable to environmental factors (e.g., sound) rather than stress. This would strengthen the conclusion that sleep parameters in the MSL are comparable to those obtained with conventional polysomnography, if sound levels are adequately controlled throughout the night.

Response: Thank you for your advice. To investigate whether the environmental factor, sound level, was associated with a decrease in %N3, we evaluated the correlation between %N3 and sound level using hourly data from N = 472 except for intervals where sleep was not occurring and sound level failed to be measured. Since %N3 was not correlated with sound level (rs = 0.10, p = 0.112 in the HSL; rs = 0.03, p = 0.665 in the MSL, S5 Figure), the decrement in %N3 in the MSL may not be explained by the influence of the sound level. We have included the following sentences in the Methods on page 15, lines 322-326: “As we found a significant increase in averaged sound in the first hour of MSL recording, we calculated the correlation between the %N3 and sound level in each hour from N = 472 data. Data intervals were removed where sleep was not occurring and sound level failed to be measured. Spearman’s correlation coefficient was used to evaluate whether the %N3 decrement was related to sound level.” We added the following descriptions to the Results on page 16, lines 365-369: “To evaluate whether sound level increment in the first hour related to %N3 decrement in the MSL, correlations between hourly averaged sound level and hourly averaged %N3 were evaluated for the HSL and MSL. No significant correlation was found between sound levels in the HSL and MSL, respectively (rs = 0.10, p = 0.112, rs = 0.03, p = 0.665, S5 Fig. A and B). ” We have added the following sentence on page 22, lines 512-514: “However, there was no correlation between %N3 and sound level, so the decrease in %N3 at the MSL may not be explained by the environmental sound factor.” We have deleted the following sentence from the Discussion: “That is, sound level fluctuation might partly relate to the reduction %N3 in MSL.”

Reviewer #1_4: Finally, there is a minor correction needed in Figure 2: the number of participants is noted as N = 16, but the final sample includes 15 participants. Please adjust or remove this notation accordingly.

Response: Thank you for your confirmation. In Fig 2A, N = 16 on the left side was corrected to N = 15. In addition, the number in each group (N = 7 and N = 8) is shown on the right side of Fig 2A. The figure caption on page 8, lines 145-147 was corrected as follows: “The participants were randomly allocated to two groups. The first group (n = 7) underwent sleep measurement at the HSL for the first two nights, and at the MSL for the next two nights; this sequence was reversed for the second group (n = 8).”

---

## [Decision Letter · Decision Letter 2]

20 Nov 2024

PONE-D-23-34836R2Mobile Sleep Lab: Comparison of polysomnographic parameters with a conventional sleep laboratoryPLOS ONE

Dear Dr. Suzuki,

Thank you for submitting your manuscript to PLOS ONE. After careful consideration, we feel that it has merit but does not fully meet PLOS ONE’s publication criteria as it currently stands. Some of the updated analyses (in response to the reviewer's previous comments lacked clarity and justification). Therefore, we invite you to submit a revised version of the manuscript that addresses the points raised during this round of review. 

We look forward to receiving your revised manuscript.

Kind regards,

Bradley R. King

Academic Editor

PLOS ONE

Journal Requirements:

Reviewers' comments:

Reviewer's Responses to Questions

**Comments to the Author**

1. If the authors have adequately addressed your comments raised in a previous round of review and you feel that this manuscript is now acceptable for publication, you may indicate that here to bypass the “Comments to the Author” section, enter your conflict of interest statement in the “Confidential to Editor” section, and submit your "Accept" recommendation.

Reviewer #1: (No Response)

2. Is the manuscript technically sound, and do the data support the conclusions?

Reviewer #1: Yes

3. Has the statistical analysis been performed appropriately and rigorously? 

Reviewer #1: Yes

4. Have the authors made all data underlying the findings in their manuscript fully available?

Reviewer #1: Yes

5. Is the manuscript presented in an intelligible fashion and written in standard English?

Reviewer #1: Yes

6. Review Comments to the Author

Reviewer #1: Thank you for addressing the previous comments and for revising the manuscript. However, I would like to ask for further clarification regarding your analysis of environmental factors, especially the sound level.

In your response, you mentioned: “In the previous manuscript, we evaluated temperature, humidity, and sound level fluctuations overnight by calculating the hourly coefficient of variations (CVs), but we thought that calculating CVs over 8 h and quantitatively evaluating them would be a more appropriate way to compare between the HSL and MSL, rather than comparing hourly CVs.” While this methodological change is acknowledged, it lacks a clear justification. Could you please elaborate on why you believe this approach is more appropriate and provide a stronger rationale for this methodological shift?

Additionally, in lines 324–326, you state: “As we found a significant increase in averaged sound in the first hour of MSL recording, we calculated the correlation between the %N3 and sound level in each hour from N = 472 data.” This implies that you still performed an hourly analysis. However, later in the Results section, the correlation is described as being calculated between hourly averaged sound levels and hourly averaged %N3, with no apparent differences between the MSL and HSL. Given that the sound level was significantly higher during the first hour of recording in the MSL, I would recommend performing a focused analysis of the correlation between %N3 and sound level specifically during the first hour of the night, rather than averaging this measure across the entire night. This approach would directly address the potential influence of the elevated sound levels observed at the start of the recording.

If you choose not to follow this recommendation, I kindly ask that you provide a detailed rationale for why the averaged measures across the night were deemed more appropriate for this analysis, despite the observed increase in sound during the first hour.

7. PLOS authors have the option to publish the peer review history of their article (what does this mean?). If published, this will include your full peer review and any attached files.

Reviewer #1: No

---

## [Author Response · Author response to Decision Letter 2]

9 Dec 2024

Response to Reviewers

Reviewers' comments:

Journal requirement_1: Please review your reference list to ensure that it is complete and correct. If you have cited papers that have been retracted, please include the rationale for doing so in the manuscript text, or remove these references and replace them with relevant current references. Any changes to the reference list should be mentioned in the rebuttal letter that accompanies your revised manuscript. If you need to cite a retracted article, indicate the article’s retracted status in the References list and also include a citation and full reference for the retraction notice.

Response: Thank you for your comments and suggestions. We have reviewed and corrected the reference list. There are no retracted articles.

Reviewer #1_1: Thank you for addressing the previous comments and for revising the manuscript. However, I would like to ask for further clarification regarding your analysis of environmental factors, especially the sound level.

In your response, you mentioned: “In the previous manuscript, we evaluated temperature, humidity, and sound level fluctuations overnight by calculating the hourly coefficient of variations (CVs), but we thought that calculating CVs over 8 h and quantitatively evaluating them would be a more appropriate way to compare between the HSL and MSL, rather than comparing hourly CVs.” While this methodological change is acknowledged, it lacks a clear justification. Could you please elaborate on why you believe this approach is more appropriate and provide a stronger rationale for this methodological shift?

Response: Thank you for your confirmation. We chose to evaluate the CVs during the 8-h sleep period because we believe that analyzing CVs over an 8-h period provides a more comprehensive reflection of overall variability during sleep compared to examining hourly changes in CVs. We considered that this analysis would more appropriately address your initial comment: "Does the level of humidity fluctuate throughout the night?" In contrast, we thought that the analysis of hourly changes in average values was suitable for addressing your comment. Therefore, we kept the analysis of averages as it was.

Additionally, in lines 324–326, you state: “As we found a significant increase in averaged sound in the first hour of MSL recording, we calculated the correlation between the %N3 and sound level in each hour from N = 472 data.” This implies that you still performed an hourly analysis. However, later in the Results section, the correlation is described as being calculated between hourly averaged sound levels and hourly averaged %N3, with no apparent differences between the MSL and HSL. Given that the sound level was significantly higher during the first hour of recording in the MSL, I would recommend performing a focused analysis of the correlation between %N3 and sound level specifically during the first hour of the night, rather than averaging this measure across the entire night. This approach would directly address the potential influence of the elevated sound levels observed at the start of the recording.

If you choose not to follow this recommendation, I kindly ask that you provide a detailed rationale for why the averaged measures across the night were deemed more appropriate for this analysis, despite the observed increase in sound during the first hour.

Response: Thank you for your valuable feedback about focusing on the first hour and seeking the relationship between %N3 and sound level. We analyzed it according to your suggestion. No correlation was seen between %N3 and noise at MSL. Thus, increasing noise during the first hour of sleep recording at the MSL may not be associated with a decrease in %N3 at the MSL. However, surprisingly, a positive correlation was found between %N3 and sound level in the first hour at the HSL. That is, %N3 increased with increasing sound level during the first hour in the HSL. The reason for this remains unclear. 

According to the suggested analysis, we added the following to the methods on page 15, lines 353-359: “As in the second night of MSL, the average hourly sound level at 1 h after lights out was higher than at 4 and 6–8 h, so we focused on the correlation between %N3 and sound level during the first hour. We predicted that the increase in sound level would be related to the decrease in %N3 in the MSL. The first and second nights were combined with the corresponding HSL and MSL to obtain the respective correlation coefficients between %N3 and sound level.”

We have added the following to the results on page 17, lines 402-407: “As an increase in sound level was observed at 1 h after lights out compared to 4 and 6–8 h in the second night of the MSL, the correlation between %N3 and sound level was examined focusing on the first hour after lights out. The results showed a positive correlation between %N3 and sound level in the HSL (rs = 0.42, p = 0.023, S5 Fig. C) and no correlation between %N3 and sound level in the MSL (rs = 0.25, p = 0.187, S5 Fig. D).”

We have added the following discussion from page 22, line 550 to page 23, line 555: “As an increased sound level might influence the decrease in %N3 in the MSL, we performed a correlation analysis. Contrary to our expectation, %N3 increased with increasing sound level during the first hour after light out in the HSL. The reason for this remains unclear. However, there was no correlation between %N3 and sound level in the MSL, so the environmental sound factor cannot explain the decrease in %N3 in the MSL.”

We have added Supplemental Information for scatter plots of %N3 and sound level in the first hour to S5 Fig. C and D.

---

## [Editor Report · Decision Letter 3]

13 Dec 2024

Mobile Sleep Lab: Comparison of polysomnographic parameters with a conventional sleep laboratory

PONE-D-23-34836R3

Dear Dr. Suzuki,

We’re pleased to inform you that your manuscript has been judged scientifically suitable for publication and will be formally accepted for publication once it meets all outstanding technical requirements.

Kind regards,

Bradley R. King

Academic Editor

PLOS ONE
---

## [Editor Report · Acceptance letter]

27 Dec 2024

PONE-D-23-34836R3 

PLOS ONE

Dear Dr. Suzuki, 

I'm pleased to inform you that your manuscript has been deemed suitable for publication in PLOS ONE. Congratulations! Your manuscript is now being handed over to our production team.

Kind regards, 

on behalf of

Dr. Bradley R. King 

Academic Editor

PLOS ONE